# Ocean-Induced Weakening of George VI Ice Shelf, West Antarctica

Ann-Sofie P. Zinck<sup>1,\*</sup>, Bert Wouters<sup>1</sup>, Franka Jesse<sup>2</sup>, and Stef Lhermitte<sup>3,1</sup>

Abstract. Channelized basal melting is a critical process influencing ice shelf weakening, as basal channels create zones of thinning and vulnerability that can potentially lead to ice shelf destabilization. In this study, we reveal and examine the rapid development of a channel within the George VI Ice Shelf's extensive channelized network, characterized by a 23 m surface lowering over a nine-year period. We study changes in ice flow, ocean circulation and heat potential as possible drivers behind the channel, under the hypotheses that it is either a fracture, a basal melt channel, or a combination of the two. Our findings show that the onset of this channel coincides with significant changes in ocean forcing, including increased ocean temperatures and salinity, that occurred during the 2015 El Niño Southern Oscillation event. Modelling of basal melting further suggests that channel re-routing has taken place over this nine-year period, with the new channel serving as a basal melt channel in the latest years. We further observe subtle shifts in ice flow indicative of fracturing. Our findings thus indicate that this channel likely contributes to the weakening of an already thin ice shelf through a combination of basal melting and fracturing. These findings offer insight into how similar potentially destabilizing processes could unfold on other Antarctic ice shelves. Monitoring the evolution of this channel and its impact on ice shelf integrity will be critical for understanding the mechanisms of ice shelf retreat, especially on heavily channelized ice shelves.

#### 1 Introduction

The future evolution of the Antarctic Ice Sheet and its potential sea level contribution in a changing climate is highly uncertain (IPCC, 2023), largely due to the uncertain response of ice shelves (Pattyn and Morlighem, 2020; Bamber et al., 2022; van de Wal et al., 2022). Ice shelves play a critical role in buttressing the ice sheet (Fürst et al., 2016) and their weakening or eventual collapse can accelerate ice flow across the grounding line, which in turn contributes to sea level rise (Pattyn and Morlighem, 2020; van de Wal et al., 2022). It is therefore crucial to understand the processes that lead to ice shelf weakening to better constrain uncertainties in sea level rise projections.

Basal melting is one such process known to weaken ice shelves (Pritchard et al., 2012; Silvano et al., 2016) and has been linked to the collapse of Wordie Ice Shelf (Doemgaard et al., 2024). Basal melting is often driven by the intrusion of Circumpolar Deep Water (CDW), a relatively warm water mass located at depth beyond the continental shelf (Silvano et al., 2016). In some regions of Antarctica, the continental shelf's bathymetry allows CDW to flow over it, enabling this warm water to reach ice shelf cavities (referred to as warm-cavity ice shelves, Silvano et al., 2016). This phenomenon occurs particularly in the

<sup>&</sup>lt;sup>1</sup>Delft University of Technology, Delft, The Netherlands

<sup>&</sup>lt;sup>2</sup>Utrecht University, Utrecht, The Netherlands

<sup>&</sup>lt;sup>3</sup>KU Leuven, Leuven, Belgium

<sup>\*</sup>Current address: University of Copenhagen, Copenhagen, Denmark

Amundsen and Bellingshausen Seas - where the Wordie Ice Shelf was located - which experience some of the highest basal melt rates on the continent (Rignot et al., 2013; Davison et al., 2023). Although basal melting is most pronounced near the grounding zone due to the pressure-dependent freezing point of seawater (Silvano et al., 2016), the resulting meltwater plume usually travels from the grounding zone towards the ice shelf front, thereby carving basal channels in the ice (Alley et al., 2022). These channels, which are more prominent in warm-cavity ice shelves (Alley et al., 2016), represent potential zones of weakness due to their reduced thickness and elevated melt rates relative to their surroundings (Alley et al., 2022).

Basal channels are dynamic features that do not necessarily remain in a steady state. They can experience changes in melt intensity, fracturing, and re-routing. In warm-cavity ice shelves, the magnitude of melting within channels and across the entire ice shelf is primarily influenced by the volume and temperature of CDW entering the ice shelf cavity (Dutrieux et al., 2014). These oceanic conditions can, in turn, be modulated by larger climatic phenomena such as the El Niño Southern Oscillation (ENSO, Huguenin et al., 2024). During El Niño events, the weakening of near-coastal easterly winds limits the shoreward supply of cooler surface waters and facilitates greater inflow of CDW onto the continental shelf, thereby intensifying basal melting (Huguenin et al., 2024). Increased availability or higher temperatures of CDW can thus lead to enhanced thinning through basal melting (Paolo et al., 2018).

Basal channels can also be associated with transverse fracturing, a phenomenon observed on several ice shelves, including Pine Island, Nansen, Moscow University, and Totten (Dow et al., 2018; Alley et al., 2024). These fractures, driven by extensional stresses within the ice (Dow et al., 2018), have triggered major calving events on Nansen and Pine Island ice shelves (Dow et al., 2018; Alley et al., 2024, 2022). Although extensional stresses are considered a primary driver, variations in meltwater availability within channels may also contribute to fracture formation (Alley et al., 2022). In addition to transverse fracturing, longitudinal fractures within basal channels have been documented (Vaughan et al., 2012; Dutrieux et al., 2013), potentially leading to full-thickness fractures and eventual ice shelf retreat (Alley et al., 2022).

40

The interactions between basal channels, fractures, and CDW-variations are highly complex, reflecting significant knowledge gaps in channel dynamics. Beyond changes in melt magnitude and fracturing, basal channels have been observed to migrate laterally or re-route entirely. Examples of such behavior include channels on the Getz (Chartrand and Howat, 2020) and Roi Baudouin (Drews et al., 2020) ice shelves, as well as other ice shelves in the Amundsen and Bellingshausen Sea sectors (Alley et al., 2024). Lateral channel migration, often in the direction favored by the Coriolis effect, has been linked to increased ocean heat availability (Alley et al., 2024). Similarly, the re-routing of a meltwater plume on Thwaites Ice Shelf has been observed to follow pre-existing fractures, underscoring the interplay between channel dynamics and ice shelf structure (Alley et al., 2024).

The importance of high-resolution basal channel melt rates on ice shelf weakening was emphasized using Basal melt rates Using the Reference Elevation Model of Antarctica (REMA) and Google Earth Engine (BURGEE, Zinck et al., 2024a). The high-resolution BURGEE melt data reveal a peculiar, and to our best knowledge so far unreported, channel on George VI Ice Shelf with high melt rates near extensive channelization (Fig. 1). The channel appears to intersect and alter other channels in the area, suggesting significant changes in the channel system during the BURGEE period (2010–2022). Given the channel's location in a heavily channelized area and its proximity to the southern ice shelf front which has been persistently retreating since 1947 (Smith et al., 2007), this channel poses a potential risk of further retreat.

**Figure 1.** A map of the study area (George VI Ice Shelf) with BURGEE basal melt rates and bathymetry from BedMachineV3 (Morlighem et al., 2020; Morlighem, 2022). The grounding line and ice shelf extent are from BedMachineV3 (Morlighem et al., 2020; Morlighem, 2022). The three profiles marked (Southern, Middle, and Northern) refer to the profiles in Fig. 6. The transect marked as "Divergence comparison" refers to the transect used for assessing the root-mean-square error of the divergence in Fig. 4. Two similar zoom-ins of the highly channelized area marked by the white square show BURGEE melt rates. The upper zoom-in further shows the approximate location of the surface depression of the old and new channels in year 2016 in teal as well as the red transect used in Fig. 3 and 4. The lower zoom-in allows for better visualisation of the melting within the channel with the new channel pointed out by the white arrow.

In this study, we examine the potential drivers behind the formation and evolution of this newly observed channel on the George VI Ice Shelf (Fig. 1). Using high-resolution surface elevation and basal melt rate data, we assess temporal changes in channel morphology. Furthermore, we explore the roles of ice flow dynamics, ocean circulation, and ocean forcing as potential contributors to channel evolution. Finally, we employ a basal melt model with realistic ocean forcing and ice shelf geometries from two time periods (2010-2016 and 2016-2022) to investigate the possibility of channel re-routing within the observed network. This multi-faceted approach aims to advance our understanding of the processes driving rapid channel formation.

## 2 Study area

Our study area focuses on the region surrounding the newly identified channel on the George VI Ice Shelf located near the southern ice shelf front, as well as the two entrances to its ice shelf cavity where CDW flows in (Fig. 1). The analysis spans the BURGEE period from 2010 to 2022.

George VI Ice Shelf is particularly vulnerable due to its already thin state and ongoing thinning (Smith et al., 2007; Davison et al., 2023). The ice shelf experiences significant melting both from its surface (de Roda Husman et al., 2024; van Wessem et al., 2023) and its base (Davison et al., 2023), making it susceptible to structural weakening. Adding to this vulnerability, studies have shown that parts of the firn layer of the ice shelf are saturated to levels that could facilitate hydrofracturing, while other areas are approaching similar thresholds with only minor increases in temperature (van Wessem et al., 2023). This dual weakening mechanism highlights the sensitivity of George VI Ice Shelf to changes in climatic conditions.

The bathymetry of the region further amplifies this vulnerability. Troughs in the continental shelf guide CDW into the cavity beneath the ice shelf (Fig. 1, Hyogo et al., 2024). Once inside the cavity, the CDW drives some of the highest basal melt rates observed in the Antarctic Peninsula (Davison et al., 2023).

#### 80 3 Data and Methods

100

In the following subsections, we describe the data and methods used to i) generate high-resolution surface elevation and basal melt rates to analyze the temporal evolution of the channel (Sect. 3.1); ii) explore potential drivers behind the channel, utilizing both ice velocity observations (Sect. 3.2) and model simulations of ocean circulation and forcing (Sect. 3.3); iii) model temporal changes in basal melt patterns near the channel driven by variations in ocean forcing and ice shelf geometry (Sect. 3.4).

## 85 3.1 Surface elevations and basal melt rates

To derive high-resolution surface elevation and basal melt rates of the George VI Ice Shelf, we use BURGEE as presented in Zinck et al. (2023) and updated in Zinck et al. (2024a). In summary, BURGEE utilizes CryoSat-2 SARin Baseline-E Level 1B radar altimetry data to co-register the 2-meter REMA digital surface model strips (Howat et al., 2022), producing high-resolution surface elevation maps. From these maps, surface elevation changes are calculated in a Lagrangian framework, from which basal melt rates are derived using information about surface mass balance, firn, and ice flow.

For this study, we use REMA strips from version s2s041 (Howat et al., 2022) instead of the older s2s030 (REMA v1) used in Zinck et al. (2023, 2024a). The latter is the only version available on the Google Earth Engine but covers only the years from 2010 to 2017, whereas the newest version is updated yearly with new data. Since George VI is frequently cloud-covered, we included these newer strips to enhance coverage. All strips were bicubically interpolated onto a 50-meter grid from their original 2-meter resolution to reduce storage requirements, both locally and on Google Earth Engine.

Surface elevations were derived by co-registering the REMA strips to CryoSat-2 radar altimetry measurements, following the method outlined in Zinck et al. (2023) with updates from Zinck et al. (2024a). First, both datasets were adjusted for dynamic and static corrections – accounting for tides, mean dynamic topography, the inverse barometer effect, and geoid referencing. Then, tilt and bias in the REMA strips were corrected by fitting a plane through the REMA/CryoSat-2 elevation residuals. Basal melt rates were calculated similarly to Zinck et al. (2024a), with a minor adjustment in the Lagrangian displacement related to the feature tracking. In previous work (Zinck et al., 2024a), strips were referenced to a median elevation map covering the period 2015-07-01 to 2017-07-01. Here, for the George VI Ice Shelf, we extended this median elevation map period to 2018-

07-01 for improved ice shelf coverage, resulting in better quality of the Lagrangian displacement. To derive the basal melt rate through the traditional mass conservation approach we used both information about firn air content from the IMAU-FDM v1.2A (Veldhuijsen et al., 2023) and surface velocities from MEaSURE ITS\_LIVE (Gardner et al., 2022) to calculate the ice flux divergence as described in Zinck et al. (2023). For the surface mass balance, however, we used a new 2 km resolution downscaled version of the regional climate model RACMO (Noël et al., 2023). The final surface elevation and basal melt maps were produced on a 50-meter grid.

#### 3.2 Ice velocities and divergences across the channel

105

To investigate ice speed and divergence across the new channel (Fig. 1), and track their temporal evolution as possible indicators of fracturing, we use monthly ice velocities from ENVEO (provided by the European Space Agency's Antarctic Ice Sheet Climate Change Initiative project). Based on Sentinel-1 (synthetic aperture radar) imagery, these velocities offer higher spatial and temporal resolution, along with greater coverage, compared to velocity products based on optical imagery feature tracking such the MEaSURE ITS\_LIVE velocities used in BURGEE. These Sentinel-1-based velocities are not used in BURGEE since they are only available from 2015 onwards and thus do not cover the entire BURGEE period.

We examine a  $\sim$ 15 km transect crossing both the persistent channel system and the new channel (Fig. 1). Yearly ice speeds are calculated by taking the median of the monthly speeds, with measurements extracted every  $\sim$ 250 meters along the transect.

To obtain the divergence along the transect, we first calculate the yearly median x- and y-components of the monthly velocity fields, then extract the values along the transect in the same manner as for ice speed. The divergence is computed by summing the velocity gradients in the x- and y-directions for the yearly velocity fields along the transect. To estimate the noise in the divergence data we calculate the root-mean-square error of the yearly divergences along a transect (Divergence comparison, Fig. 1) where rifting and thus changes in the divergence field are not expected to occur.

Finally, to investigate the possibility of rifting in a greater spatial detail, yearly effective strain rates of the region are calculated based on the yearly median x- and y-components of the velocity fields. The horizontal strain rate components are calculated using a local quadratic regression of  $2 \text{ km} \times 2 \text{ km}$ , following the same procedure as in De Rydt et al. (2018).

# 3.3 Ocean heat and circulation

To investigate temporal changes in ocean heat content during the study period, we use output from the Amundsen-Bellingshausen Seas regional configuration of the Massachusetts Institute of Technology general circulation model (MITgcm), which has been validated against observational oceanographic data (Hyogo et al., 2024; Park et al., 2024). This model configuration, originally developed by Nakayama et al. (2018), features a horizontal grid spacing of 2–4 km and 70 vertical layers. The vertical resolution varies, with layer thicknesses ranging from 10 meters near the surface to 450 meters in the deepest regions ( $\sim$  -6000 m), and between 70–90 meters at depths of -500 to -1000 meters. The model uses constant ice shelf geometry and ocean bathymetry from BEDMAP-2 (Fretwell et al., 2013) and has a monthly temporal output. We examine the model's monthly output of temperature and salinity for all years between 2010 and 2020 to determine if shifts in ocean heat potential can explain the channel's development. Temperature and salinity profiles are extracted from three cross-sections: the northern and south-

ern entrances to the ice shelf cavity, and near the channel (Fig. 1). Even though the new channel is located near the southern entrance, CDW entering through the northern entrance might contribute to the meltwater plume within the channel system, due to the sub-shelf circulation (Hyogo et al., 2024). Anomalies in temperature and salinity are calculated by subtracting the 2010 to 2020 time-averaged values at each depth level. Furthermore, we compute the depth-averaged monthly temperature and salinity for the water mass below -300 m, where CDW is found in the George VI cavity (Jenkins and Jacobs, 2008; Holland et al., 2010).

## 3.4 Modelling of basal melt rates

We use the two-dimensional basal melt model LADDIE (Lambert et al., 2023) to simulate basal melt rates with greater spatial detail (compared to the MITgcm output) for George VI Ice Shelf in two scenarios corresponding to before and after the new channel emerged: i) January 2010 to July 2016 (pre-emergence experiment), and ii) July 2016 to December 2022 (post-emergence experiment). LADDIE solves the vertically integrated Navier-Stokes equations to compute the temperature, salinity, thickness, and horizontal velocities of the meltwater plume below the ice shelf. Basal melt rates are calculated using the three-equation formulation for melting and refreezing (Holland and Jenkins, 1999; Jenkins et al., 2010), which includes conservation of heat and salt, along with a constraint that keeps the ice-ocean interface at the local freezing point. As LADDIE is designed to simulate steady-state melt patterns for a given geometry and ambient ocean forcing, we compare the pre- and post-emergence states in a "snapshot" manner, prescribing two fixed ocean forcing profiles representative of the respective phases. The goal is to investigate potential changes in plume direction and melt patterns near the channel, using two different ice shelf geometries and ocean forcings, resembling the two different time periods, while keeping other model parameters constant. Below, we describe how these ice shelf geometries and ocean forcings are derived, and how the model is tuned to match observed basal melt rates.

#### 3.4.1 Ice shelf geometry

The ice shelf geometries for the pre-emergence and post-emergence scenarios are generated using a combination of BedMachineV3 (Morlighem et al., 2020; Morlighem, 2022) data and co-registered REMA strips (Sect. 3.1). BedMachineV3 provides bathymetry, ice shelf mask, and the ratio between surface elevation and ice shelf thickness, which we use to transform REMA-derived surface elevations into ice thickness and draft.

Following the BURGEE approach, we construct two high-resolution surface elevation maps. This involves first aligning REMA strips horizontally and then vertically with respect to BedMachineV3. Because the REMA strips originate from different times, we displace them to common reference dates: 2013-01-01 for the pre-emergence map and 2023-01-01 for the post-emergence map. Horizontal displacements are applied using MEaSURES ITS\_LIVE velocities (Gardner et al., 2022). For the pre-emergence geometry, we use strips acquired up to 2016-07-01, while the post-emergence geometry includes strips from 2016-07-01 onwards. Since velocity-based displacement only provides approximate alignment, we further adjust the strips via feature tracking relative to median elevation maps (2012-07-01 to 2015-07-01 for pre-emergence; 2018-07-01 to 2022-12-31 for post-emergence).

To be able to use the BedMachineV3 surface-elevation-to-thickness ratio, vertical alignment is required because REMA and BedMachineV3 are referenced differently, due to differing dynamic and static corrections. As in BURGEE, we correct for this by fitting a plane to the elevation residuals between REMA strips and BedMachineV3. Final surface elevation maps are then obtained by taking the median of the corrected strips (for each time period) and interpolating them onto the 500 m BedMachineV3 grid.

Ice thickness for both geometries is derived using BedMachineV3's surface-elevation-to-thickness ratio. However, because the post-emergence surface map uses more recent data (2016–2023) than BedMachineV3 (2010–2017), we replace grounding-zone elevations (within ~1.5 km) with those from the pre-emergence map to reduce inconsistencies. Finally, all negative thickness values, unrealistic ice shelf thicknesses (>2000 m), and remaining gaps are replaced with BedMachineV3 values in both geometries.

# 3.4.2 Ocean forcing

We force LADDIE with a 1D temperature and salinity profile (Lambert et al., 2023) based on MITgcm results from 2010 and 2020 (Hyogo et al., 2024), for the pre-emergence and post-emergence experiment, respectively. We average the May-August MITgcm temperatures and salinities from the Northern and Southern profiles (Fig. 1) to obtain the average forcing for 2010 and 2020 (Fig. 2). We only consider austral winter months (May-August) to reduce the noise level from the seasonality in the upper ocean layers. To avoid sudden changes in temperature and salinity present in the MITgcm outputs and to ensure a stable stratification, we describe both salinity and temperature as a tangent hyperbolic function. For the temperature, the tangent hyperbolic function is already built into LADDIE, where the surface temperature ( $T_0$ ) is set to the surface freezing point based on the surface salinity ( $S_0$ ) following

$$T_0 = l_1 S_0 + l_2. (1)$$

Here  $l_1$  and  $l_2$  are the freezing point salinity coefficient and the freezing point offset, respectively. The temperature (T) as a function of depth (z) is then given by

$$T(z) = T_1 + (T_0 - T_1) \frac{1 + \tanh\left(\frac{z - z_0}{z_1}\right)}{2},\tag{2}$$

where  $T_1$  is the temperature at depth,  $z_0$  is the reference depth for the thermocline, and  $z_1$  is the scaling factor of the thermocline, which determines the thermocline gradient. We use a similar tangent hyperbolic function for the salinity (S), following

$$S(z) = S_1 + (S_0 - S_1) \frac{1 + \tanh\left(\frac{z - z_{0,S}}{z_{1,S}}\right)}{2}.$$
(3)

Here,  $S_1$  is the salinity at depth,  $z_{0,S}$  is the halocline reference depth (as opposed to the thermocline for the temperature), and  $z_{1,S}$  is the halocline scaling factor. Surface salinity, salinity at depth, and temperature at depth are all roughly based on the MITgcm profiles, whereas the thermocline/halocline depths and scaling factors are tuned to match the MITgcm profiles.

**Figure 2.** Average temperature (**a**) and salinity (**b**) profiles from winter months (May-August) of MITgcm in 2010 (dashed blue) and 2020 (dashed red), alongside with the tangent hyperbolic prescribed temperature (**a**) and salinity (**b**) used in LADDIE for the pre-emergence run (solid blue) and post-emergence run (solid red).

**Table 1.** Forcing parameters used in the pre-emergence and post-emergence experiments.

| Parameter                                                    | pre-emergence | post-emergence |
|--------------------------------------------------------------|---------------|----------------|
| Temperature at depth, $T_1$ [°C]                             | 1.50          | 1.65           |
| Thermocline depth, $z_0$ [m]                                 | -150          | -110           |
| Thermocline scaling factor, $z_1$ [m]                        | 150           | 150            |
| Freezing point salinity coef., $l_1$ [°C psu <sup>-1</sup> ] | 3.733e-5      | 3.733e-5       |
| Freezing point offset, $l_2$ [°C]                            | 8.32e-2       | 8.32e-2        |
| Surface salinity, $S_0$ [psu]                                | 33.30         | 33.10          |
| Salinity at depth, $S_1$ [psu]                               | 34.65         | 34.65          |
| Halocline depth, $z_{0,S}$ [m]                               | -100          | -100           |
| Halocline scaling factor, $z_{1,S}$ [m]                      | 90            | 90             |

For both temperature and salinity the tangent hyperbolic only starts to diverge from the MITgcm profiles at depths above ~-100 m (Fig. 2), which is shallower than the ice shelf draft in most parts of the ice shelf with the exception of a few areas in the northern part of the ice shelf. All forcing parameters for the pre-emergence and post-emergence experiments are tabulated in Tab. 1.

**Table 2.** LADDIE parameters used in both experiments.

| Parameter                                              | Value               |
|--------------------------------------------------------|---------------------|
| Time step [s]                                          | 36                  |
| Horizontal resolution [m]                              | 500                 |
| Equilibrium time [model days]                          | 60                  |
| Top drag coefficient, $C_{d,top}$                      | $3.0 \cdot 10^{-4}$ |
| Minimum thickness, $D_{min}$ [m]                       | 2                   |
| Internal ice temperature [°C]                          | -25                 |
| Tidal velocity [m s <sup>-1</sup> ]                    | 0.01                |
| Horizontal viscosity [m <sup>2</sup> s <sup>-1</sup> ] | 6                   |
| Horizontal diffusivity $[m^2 s^{-1}]$                  | 1                   |
|                                                        |                     |

## **3.4.3** Tuning

LADDIE has two tuning parameters; the minimum meltwater layer thickness  $(D_{min})$  and the drag coefficient  $(C_{d,top})$  applied to the friction velocity in the basal melting formulation (Lambert et al., 2023). We use the latter as the main tuning parameter due to its direct influence on basal melt rates as tuning  $C_{d,top}$  roughly corresponds to scaling the basal melting magnitude up and down. To calibrate the model, we iteratively adjust  $C_{d,top}$  to approximate the maximum BURGEE melt rates observed near the channel of interest using the pre-emergence geometry. Once determined, this value of  $C_{d,top}$  remains fixed across the pre-emergence and post-emergence experiments, allowing us to focus on the effects of changing geometry and forcing. The full list of model parameters specific for our experiments is provided in Tab. 2. Appendix A1 and A2 contains sensitivity tests of the internal ice temperature and  $C_{d,top}$ , respectively, and Appendix A3 contains a control-run of the impact of the geometry versus forcing on the modelled melt rates.

# 4 Results

The basal melt rate from 2010 to 2022, shown in Fig. 1, reveals a general pattern consistent with previous studies (Davison et al., 2023): higher melt rates are observed in the southern region of the ice shelf compared to the north. Furthermore, as is typical for most ice shelves, the highest melt rates are concentrated near the grounding zone, with channels extending from these zones across mainly the southern part of the ice shelf. A closer examination of the newly identified channel highlights its position within a densely channelized area. This channel stands out with exceptionally high melt rates, reaching up to approximately  $30 \text{ m yr}^{-1}$ , which surpasses the melt rates in the surrounding channels. The channel appears to intersect existing channels, complicating the interpretation of the meltwater plume's pathway.

By examining the surface elevations over the study period, it becomes evident that the new channel has developed during this period; it began forming around 2015, reflected by a narrow surface depression in Fig. 3d. This depression deepens progres-

**Figure 3.** (a-i) surface elevations through time from 2012/13 in (a) to 2022/23 in (i). In panel e the old/persistent and new channel are marked by red dots. The missing years indicated in the legend are years without REMA coverage in this area. (j) shows the surface elevations in a Lagrangian framework along the transect marked and lettered for direction in (e) which crosses both the persistent channel (surface depression deepest at around 8.2 km) and the new channel (around 11 km). The sudden dip around 1 km in one of the surface elevations is caused by a few contaminated pixels in that given REMA strip.

sively throughout the study period by 23 m between 2013/14 and 2022/2023, with two further key observations emerging. First, as the depression associated with the channel becomes more pronounced (Fig. 3j, around 11 km), the deepest part of the older channel becomes shallower (Fig. 3j, around 8 km), while the flanks of the older channel are lowering (Fig. 3j, around 4-8 km and 9-10 km), thereby widening this pre-existing channel. These flanks are also associated with high melt rates ( $\sim 15 \text{ m yr}^{-1}$ , Fig. 1), as opposed to the deepest part of the channel ( $\sim 0 \text{ m yr}^{-1}$ , Fig. 1). The continuous lowering of the flanks suggests that the channel system is not in a steady state, with the closure of the deepest part possibly indicating channel re-routing, where the new channel may now serve as the primary basal melt pathway. Second, just downstream of the channel (Fig. 3j, around 12 km) a slight surface elevation bump appears from 2016/17 onward. This type of bump, known as flanking uplift, is typically associated with fractures on ice shelves (Walker and Gardner, 2019). However, narrower channels can cause similar flanking uplift (Stubblefield et al., 2023). These findings suggest that the channel could either be a fracture, a basal melt channel, or a combination of both (i.e., a fracture that serves as a melt plume pathway or a channel that has begun to fracture). Optical imagery throughout the study period from Landsat 8 and Sentinel 2 also do not provide any clear answer as to whether it is a channel or fracture (Suppl. Movie). Since both basal channels and fractures can weaken ice shelves (Alley et al., 2022, 2024), the presence of either on George VI, a relatively thin ice shelf with a relatively warm atmosphere, likely contributes to its weakening (Smith et al., 2007).

To further investigate fracturing as a possible driver behind such a fast developing channel we explore changes in ice flow across the channel. Ice shelf fracturing is typically linked to variations in ice speed and divergence caused by stretching. Our analysis of ice speed and divergence across the channel show speeds along the transect fluctuating substantially in both 2014/15 (minimum: 372 m yr<sup>-1</sup>, maximum: 395 m yr<sup>-1</sup>; Fig. 4a) and 2015/16 (minimum: 375 m yr<sup>-1</sup>, maximum: 397 m yr<sup>-1</sup>; Fig. 4b) with an isolated peak in the ice speed at the channel location in 2015/16. That peak is associated with sudden changes in divergence from -0.02 yr<sup>-1</sup> (compression) just upstream of the channel to 0.03 yr<sup>-1</sup> (stretching) just downstream of the channel (10-11 km, Fig. 4b), which could indicate fracturing as a potential driver of the channel. These changes, however, are rather subtle in comparison to the noise level and to observations across wider fractures ( $\sim$  1-4 km) on e.g. Ross Ice Shelf, where divergences reaches 30-80 m yr<sup>-1</sup> at their maximum (Walker and Gardner, 2019). The same pattern of a noisy divergence signal in the first few years followed by a limited divergence signal near the channel in the later years also becomes obvious from the spatial analysis of effective strain rates (Fig. 5). The low magnitude of the changes could potentially be due to the coarse resolution of the velocity product (200 m) relative to the channel's width and depth. Finally, in the later years (2017 and 2020, Fig. 4c and d), both ice speeds and divergences are more stable, without any outstanding signals in the vicinity of the channel.

Focusing on changes in ocean heat as possible driver of channel changes, we investigate temporal changes in ocean temperature and salinity. Figure 6a-f illustrate temperature and salinity anomalies for the Northern, Middle, and Southern profiles (Fig. 1), revealing a regime shift from cold and fresh conditions to warmer and saltier conditions below the surface layer across all profiles. At the Northern profile, this shift begins around 2013, intensifying in 2015/16, after which sub-shelf temperatures remain above the temporal mean. Similar, albeit weaker, trends are observed in the Middle and Southern profiles. In the deeper ocean layers below -300 m, where CDW resides, both average temperature and salinity have increased over the study period

**Figure 4.** Surface elevation (same coloring as in Figure 3j), ice speed (blue) and divergence (green), all in a Eulerian framework, along the transect marked in Figure 3e. The dashed green line indicate zero divergence, to easier distinguish between stretching (positive values) and compression (negative values). The shaded green area indicates the noise in the signal as described in Sect. 3.2. The dashed black line marks the approximate location of the channel. (a) is year 2014/15, (b) is 2015/16, (c) is 2016/17, and (d) is 2019/20.

**Figure 5.** Effective strain rates from 2015-2021 (**a-g**) with the position of the original and new channel from 2012/13 and 2021/22, respectively, marked in all panels.

across all three profiles (Fig. 6g-i). Notably, a jump in salinity is observed around 2016, with the most pronounced increase in the Northern profile (Fig. 6g), where the strongest temperature and salinity anomaly shifts also occur (Fig. 6a and d), although the average temperature at depth already starts increasing in 2011 (Fig. 6g). This shift in ocean regime aligns with the 2015 El Niño Southern Oscillation (ENSO) event, with its effects already having been shown to reach George VI (Boxall et al., 2024). These changes in oceanic conditions, partially driven by ENSO, thus indicate more heat available for basal melting, likely intensifying and accelerating the meltwater plume, which in turn may cause higher melt rates and alterations in the plume pathway, influenced by the ice shelf's evolving geometry. Furthermore, in the Southern temperature anomaly profile, where several melt channels have their outflows (Fig. 1), the upper ocean layers have become fresher since 2016 (Fig. 6f), possibly indicating increased meltwater outflow.

The increased meltwater outflow is supported by MITgcm-derived ice-shelf-wide and channel-area mean basal melt rates showing an increase between 2010 and 2020 (Fig. 7a), as does the mean melt rate within the channel area (Fig. 7b), thus supporting the increased meltwater outflow. The time series in Fig. 7 also indicate that the 2015 ENSO event coincides with an increase in the maximum ice-shelf-wide melt rate, and that the end of this ENSO event correlates with a shift towards higher mean melt rates both across the entire ice shelf and within the channel area. This shift also coincides with the onset of the most rapid deepening of the new channel (Fig. 7c). It is important to note that MITgcm uses a static ice-shelf geometry, which implies that changes in modelled basal melting do not reflect shifts in melt because of changing channel-geometry. However, for a new channel to arise or to modify an existing channel, elevated melt rates are likely needed to give the plume extra energy to cut new pathways. The amplified MITgcm melt rates after the 2015 ENSO event may therefore be indicative of more energy available for the plume to create new pathways.

**Figure 6.** MITgcm ocean temperature (**a-c**) and salinity (**d-f**) anomalies of the Northern profile (**a** and **d**), Middle profile (**b** and **e**), and Southern profile (**c** and **f**) as marked in Figure 1. Panel **g-i** show the average temperature and salinity at all depths 

**Figure 7.** Time series of (**a**) MITgcm mean (black) and maximum (red) basal melt rate of the entire ice shelf, (**b**) MITgcm mean (black) and maximum (red) basal melt rate of the channel-area, (**c**) REMA channel elevation (black) and REMA Lagrangian elevation change (red) of the new channel as marked in Fig. 3, and (**d**) oceanic Niño index from NOAA (Huang et al., 2017) of which values above 0.5 °C in at least 5 consecutive instances indicates an ENSO event (marked as red shadings in the curve in (**d**) and as vertical red-shaded areas in (**a-c**)).

along the sides of the channel, similar to BURGEE, which explains the widening of the old channel seen in Fig. 3j. Notably, the observed BURGEE melt pattern (Fig. 8g) near the channel appears to combine elements from both the pre-emergence and post-emergence experiments, with channels following the "original" channel system as in the pre-emergence scenario combined with high melt rates within the new channel as in the post-emergence scenario. This agreement between observations and models suggests that the plume pathway may have shifted during the BURGEE observational period, now following a new route as indicated by the post-emergence experiment.

**Figure 8.** (a-c) Ice shelf draft, LADDIE plume velocity, and LADDIE basal melt rate of the pre-emergence experiment. (d-f) Same as (a-c) but for the post-emergence experiment. Arrows on (a-b) and (d-e) are the plume velocities. The dashed green line in (c) marks the location of the original channel in 2012/13 and the dashed green line (f) marks the location of the new channel in 2021/22. (g) BURGEE basal melt rates.

#### 5 Discussion

In this study, we uncovered a new channel within the channelized basal melting network of the George VI Ice Shelf, characterized by rapid changes in surface elevation, from 48 m in 2013/14 to 25 m in 2022/23. The onset of this channel coincides with both a shift towards a warmer ocean regime and increased modelled melt rates as well as subtle divergence changes across the channel, all aligning with the 2015 ENSO event (Boxall et al., 2024). ENSO has already been linked to an intensified inflow of CDW onto the continental shelf during El Niño years (Huguenin et al., 2024), supporting the MITgcm results of a warmer regime and increased basal melt rates after 2015. The 2015 El Niño also stands out in terms of its atmospheric and surface impacts: in the George VI catchment it produced the strongest positive surface mass balance anomalies observed in recent decades, while continent-wide SMB changes during spring extended well beyond the range of background variability and penetrated further inland than during earlier extreme events (Macha et al., 2025). The same event has further been linked to the acceleration of glaciers feeding into the George VI Ice Shelf (Boxall et al., 2024). Additionally, ENSO events have been shown to enhance basal melting on ice shelves in the Amundsen Sea Sector (Paolo et al., 2018). While we cannot definitively conclude that ENSO caused the appearance of this channel, the timing is notable and the potential link to ENSO is important, as the projected increase in El Niño frequency suggests even further acceleration of ice-shelf basal melting in Antarctica in the future (Cai et al., 2023).

The observed new channel could represent a basal melt channel, a fracture, or a combination of both. Our investigations of the channel's origin point towards either a basal melt channel or a combination of both. The uplift in the surface near

the depression, coupled with signs of stretching in the divergence field downstream of the channel in 2015/16, could suggest fracturing. However, these signals are subtle, leaving us unable to confirm or dismiss the possibility of fracturing with certainty. Furthermore, it should be noted that channels themselves can generate viscous flow independent of any fracturing (Wearing et al., 2021), which implies that both the observed flanking uplift as well as the subtle divergence signal could also purely be due to channelization. At the same time, changes in ocean temperature, salinity, and circulation point to evolving and strengthening basal melt conditions that could encourage channel re-routing. Our LADDIE modelling results further support this possibility, indicating that even if the channel originated as a fracture, the meltwater plume may now be using it as a new pathway, which could further deepen the fracture, provided the initial fracture has a basal component. The new channel is narrow, whereas LADDIE has been primarily validated in broader channels with gently sloping sidewalls (Lambert et al., 2023). Flow in narrow, steep-sided channels may involve additional complexity, including ageostrophic processes (Millgate et al., 2013), that may not be fully captured in our setup. While we consider our current configuration sufficient for assessing plume pathways, caution is warranted when interpreting the exact melt rates and plume speeds in the channel.

To assess the potential impact of the new channel on the weakening of George VI Ice Shelf, we can draw parallels to other ice shelves where interactions between basal melt channels and fractures have led to structural instability. A notable example is Pine Island Ice Shelf, where a channel first observed in the 1970s progressively thinned and extended over time (Alley et al., 2022, 2024). This channel triggered both transverse fractures and fractures along the channel's length, eventually leading to calving and retreat along the channel from approximately 2018 to 2022 (Alley et al., 2022, 2024). The new channel on George VI Ice Shelf is located just about 30 km from the southern edge, raising concerns that continued melting, thinning, and weakening in this region could drive significant structural changes. These changes may include enhanced fracturing, calving, and ultimately, retreat of the ice shelf in this highly channelized area. Given projections of increased ocean heat availability in the future (Naughten et al., 2023), basal melting is expected to intensify, further amplifying the vulnerability of this portion of George VI Ice Shelf.

To definitively determine whether the channel is a fracture, a basal melt channel, or a combination of the two, field measurements, including basal melt rates, ice deformation monitoring, and radar surveys of ice thickness, are necessary. In parallel, a high-resolution, fully coupled model capable of simulating both ice flow, fracturing, and basal melting would be required to rigorously evaluate the interactions between channels and fractures and to assess causality in their formation. While our current results cannot unambiguously establish the mechanisms behind this channel, they provide a direct observation of weakening in a highly channelized, vulnerable portion of the ice shelf. These results underscore how quickly ice shelf channels - important for ice shelf integrity - can occur and how easily small-scale changes might go unnoticed. Closely observing the continued evolution of the ice shelf and its integrity will be crucial in understanding these weakening processes. Such knowledge could also be valuable for other heavily channelized ice shelves, like Pine Island and Totten, which both have a higher projected potential sea level rise contribution (Seroussi et al., 2020, 2023).

#### 6 Conclusions

355

Our study highlights the rapid emergence of a significant channel on the George VI Ice Shelf, marked by a 23 m surface lowering over just nine years. The appearance of the channel aligns with both changes in ocean forcing, most notably increased ocean temperatures and salinity, and subtle changes in ice divergence, both of which coincide with the timing of a major ENSO event. While the exact link between ENSO and the development of this channel remains speculative, the temporal correlation suggests that large-scale climate patterns may have a role in amplifying basal melting and possibly in re-routing meltwater pathways on Antarctic ice shelves.

The presence of such a fast-evolving channel on an already thin and vulnerable ice shelf like George VI likely has destabilizing effects, accelerating its weakening through both basal melting and fracturing. The behavior of this channel on George VI may offer valuable insights into how sudden changes in ocean forcing could trigger similar destabilizing processes elsewhere.

Moving forward, continuous monitoring of this channel and its evolving impact on George VI is crucial. The lessons learned from tracking its development may provide critical information on the future behavior of other highly channelized ice shelves undergoing changes in ocean conditions. Understanding these processes is essential for better projecting potential ice shelf retreat and the associated contributions to global sea-level rise.

#### 360 Appendix A: LADDIE sensitivity tests

## A1 Ice temperature

Measurements of internal ice temperatures on George VI Ice Shelf are scarce, making it difficult to select an appropriate value for use in models. To assess the sensitivity of modelled LADDIE basal melt rates to the choice of internal ice temperature, we ran two additional LADDIE experiments with a higher internal ice temperature (-10 °C) and compared them with the original experiments, which used -25 °C (Fig. A1). The results indicate that internal ice temperature has only a minor influence on modelled basal melt rates, with higher temperatures producing slightly greater melt. The effect on the plume pathway is negligible. We can therefore have good confidence in our modelled LADDIE results, as the purpose of using LADDIE is to assess whether the new channel can function as a basal channel.

# A2 Top drag coefficient

The top drag coefficient used in this study (3e<sup>-4</sup>) is lower than the typical range of 1e<sup>-3</sup> – 3e<sup>-3</sup> commonly employed in ice sheet models (Jourdain et al., 2017; Mathiot et al., 2017; Rosevear et al., 2022). A reduced value of C<sub>d,top</sub> may give greater weight to plume temperature relative to plume velocity in determining melt rates. We adopted this lower value because it was necessary to match the integrated melt. However, we suspect this discrepancy arises from the direct extrapolation of ice shelf front temperature profiles (Northern and Southern profiles, Fig. 1) into sub-shelf cavity conditions, where temperatures may in fact be colder and the thermocline deeper than currently assumed (Bett et al., 2024).

**Figure A1.** (a-c) Ice shelf draft, LADDIE plume velocity, and LADDIE basal melt rate of the pre-emergence experiment with an internal ice temperature of -25 °C. (d-f) same as (a-c) but for an ice temperature of -10 °C. (g-i) Ice shelf draft, LADDIE plume velocity, and LADDIE basal melt rate of the post-emergence experiment with an internal ice temperature of -25 °C. (j-l) same as (g-i) but for an ice temperature of -10 °C.

To investigate this, we perform a sensitivity test of the drag coefficient's effect on modelled melt rates, in combination with adjusted ocean forcing. First, we extract temperature and salinity from a MITgcm profile located within the sub-shelf cavity in an area of deep bathymetry (Deep bathymetry profile in Fig. A2c). These data are used to update the tangent hyperbolic prescribed temperature and salinity in LADDIE (Fig. A2a–b). The updated profiles have a thermocline (halocline) depth of -200 m (-200 m) and a scaling factor of 75 m (70 m), representing a deeper and sharper thermocline/halocline than in the original experiments. The temperature profile shown in Fig. A2a has a temperature at depth of 1.28 °C, based on the 2010 MITgcm Deep bathymetry profile.

For both the pre- and post-emergence geometries, we conduct three  $C_{d,top}$  sensitivity experiments and compare them with the original simulations. Each sensitivity experiment uses a drag coefficient of  $1 \times 10^{-3}$  but different temperatures at depth: 0.8 °C, 1.0 °C, and 1.28 °C, respectively, to explore a reasonable range of expected values and assess temperature sensitivity, recognizing that MITgcm may not perfectly represent reality.

The  $C_{d,top}$  sensitivity experiments for both the pre- (Fig. A3) and post-emergence (Fig. A4) geometries show that the main effect is on the magnitude of melting, with only minor changes in plume pathways. The differences in plume pathways involve a slightly more focused plume, particularly in the upper-left region of the plot, more closely resembling the sharper pattern

Figure A2. Average temperature (a) and salinity (b) of the Southern and Northern profiles combined from winter months (May-August) of MITgcm in 2010 (dotted blue) and 2020 (dotted red), alongside with average MITgcm winter months (May-August) temperature (a) and salinity (b) of the Deep bathymetry profile marked in (c) from 2010 (dashed blue) and 2020 (dashed red), and the tangent hyperbolic prescribed temperature (a) and salinity (b) used in the LADDIE  $C_{d,top}$  sensitivity tests (here shown with a temperature at depth of 1.28 °C). (c) shows BedMachineV3 bathymetry alongside with forcing profile locations.

seen in the observed melt rates. Also, enhanced  $C_{d,top}$  results in the flow being concentrated more through the new channel than through the old channel.

These sensitivity tests confirm that the main conclusion from the LADDIE simulations remain valid: the new ice-shelf geometry enables basal meltwater to flow through the newly formed fracture or channel.

**Figure A3.** (a-c) Ice shelf draft, LADDIE plume velocity, and LADDIE basal melt rate of the pre-emergence experiment. (d-f), (g-i), and (j-l) same as (a-c) but with a top drag coefficient of  $1e^{-3}$  and a temperature at depth of 1.28 °C, 1.0 °C, and 0.8 °C, respectively.

**Figure A4.** (a-c) Ice shelf draft, LADDIE plume velocity, and LADDIE basal melt rate of the post-emergence experiment. (d-f), (g-i), and (j-l) same as (a-c) but with a top drag coefficient of  $1e^{-3}$  and a temperature at depth of 1.28 °C, 1.0 °C, and 0.8 °C, respectively.

Figure A5. (a-c) Ice shelf draft, LADDIE plume velocity, and LADDIE basal melt rate of the pre-emergence experiment. (d-f) same as (a-c) but using ocean forcing from the warmer post-emergence experiment. (g-i) Ice shelf draft, LADDIE plume velocity, and LADDIE basal melt rate of the post-emergence experiment. (j-l) same as (g-i) but using ocean forcing from the colder pre-emergence experiment.

# A3 Forcing vs geometry

To investigate whether the forcing or geometry has the most impact on the modelled LADDIE malt rates we perform a "control" run, where we force the pre-emergence geometry with the warmer post-emergence forcing and vice-versa. The results of this control run (Fig. A5), show - not surprisingly - that the geometry is the main driver, and that the forcing just impacts the magnitude of the melting.

Code and data availability. The BURGEE code is publicly available at https://github.com/aszinck/BURGEE (Zinck, 2023), likewise is LADDIE (https://github.com/erwinlambert/laddie). The derived melt rates as well as surface elevations are also publicly available (Zinck et al., 2024b). The REMA strips are available from the Polar Geosptaital Center (https://www.pgc.umn.edu/data/rema/) and CryoSat-2 data is available from the European Space Agency (https://earth.esa.int/eogateway/documents/20142/37627/CryoSat-Bas eline-D-Product-Handbook.pdf). BedMachine V3 is available from NASA National Snow and Ice Data Center (https://nsidc.org/data/NSIDC-0756/versions/3) and MEaSUREs ITS\_LIVE velocities are available from https://doi.org/10.5067/6II6VW8LLWJ7 and https://nsidc.org/data/NSIDC-0756/versions/3. The ENVEO monthly velocities provided by the European Space Agency's Antarctic Ice Sheet Climate Change Initiative project are available from https://cryoportal.enveo.at/data/. The regional MITgcm model output is available from https://ecco.jpl.nasa.gov/drive/files/ECCO2/LLC1080\_REG\_AMS/Hyogo\_et\_al\_2022 (Hyogo et al., 2024).

*Author contributions*. The study was designed by ASPZ, BW, and SL and carried out by ASPZ. FJ made significant contributions in setting up the LADDIE experiments and interpreting the results therefrom. ASPZ wrote the paper with input from all authors.

Competing interests. BW and SL are members of the editorial board of The Cryosphere. The authors declare no further competing interests.

Acknowledgements. This study is part of the HiRISE project funded by the Dutch Research Council (NWO, no. OCENW.GROOT.2019.091). The authors would like to thank Shuntaro Hyogo for providing guidance on how to read the MITgcm data and Yoshihiro Nakayama for providing guidance on how interpret the MITgcm data.

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
