# Peer review of "Ocean-Induced Weakening of George VI Ice Shelf, West Antarctica"

_EGUsphere, 2025_

## Referee Comment (RC1)

**General comments:**

In this study, the authors uncovered a newborn basal channel within the extensive channelized network of the southern frontal area of George VI Ice Shelf. That channel exhibits remarkably higher basal melt rate than the surrounding area, and may have the potential to influence the ice shelf weakening. The authors argue that the emergence of that channel is likely linked to the 2015 ENSO event. I think the manuscript is well written and structured, and the uncovering of the new channel is intriguing, but I also do think the mechanism behind the generation of that channel and its influence on the ice-shelf instability are poorly demonstrated by the oceanic models the authors used in view of the quite inadequate horizontal resolution, fixed thermohaline forcing, inconsistence between the model parameter and the modeled results, and the insignificant thermal and speed differences between the BEFORE and AFTER experiments, etc. Please refer to my major comments for more details. As such, I regret that I could not support to publish this manuscript in its current form in The Cryosphere. Some minor suggestions are also provided.

**Major comments:**

First of all, the adopted horizontal spacing in LADDIE is 500 m, and that in the previous MITgcm-based simulation the authors utilized is much coarser, that is, 2-4 km. In contrast, the width of the new channel unveiled here is rather narrow, that is, only 2 km (as shown in Fig. 4j). That means such a fine basal topography is not represented at all in the utilized MITgcm model, and is also poorly represented in the LADDIE. Therefore, those coarse horizontal resolutions are far from adequate to resolve the interaction between the new channel and the underlying ocean. Second, as shown in Fig. 6b and h and Fig. 7c and f, the thermal and speed differences just below the ice base of the focused area between the BEFORE and AFTER periods are no more than 0.2 °C and 2 cm/s, respectively. Those small anomalies I think would not account for "the rapid emergence of a significant channel" with sufficient credibility. Third, the adoption of the constant (only winter-averaged in 2010 and 2020) profiles to force LADDIE seems to be some inappropriate in view of the fast-evolving southern part of George VI Ice Shelf as illustrated in this study. Fourth, there is a significant inconsistence between the large channelized basal melt rate (up to 30 m s$^{-1}$ as shown in Fig. 1) and both low drag coefficient ($3\times10^{-4}$) and plume speed (the maximum is only 0.15 m/s as shown in Fig. 8b and e), which would make me question the simulated results of LADDIE. At last, the LADDIE results shown in Fig.

8, for me, do not provide any new insights into the channelized meltwater plume and the corresponding basal melting. The plume tends to be topographically guided by the basal channels, and the basal melting is amplified therein. That finding has been proved in a wealth of literatures as listed at the end of this report.

Therefore, after reviewing this manuscript I still have no idea why that channel was born, and what it will bring about for the upcoming evolution of George VI Ice Shelf. I think that could not be simply attributed to ocean warming, the link of which to ENSO, however, remains unclear (as stated in Line 277). Although the authors have acknowledged that "there is a disparity in scale between the model grid size and the size of the channel (Line 255)" and "interpreting circulation changes in more detail and on a smaller scale, particularly in the immediate vicinity of the channel, remains challenging given the coarse model resolution (Line 257-258)", it is the model flaws they mentioned above that are **SO** critical to resolve my concern. In that sense, maybe the coupled ice sheet-ocean models (as used in Gladish et al., 2012 and Sergienko, 2013) but with sufficiently high resolution would be an effective way.

**Specific comments:**

The title: I think the temporal scale of this weakening should be specified.

Line 16: "…, largely due to the unknown response of ice shelves."_response to what? From my point of view, the response of ice shelves to the changing oceanic and climatic conditions is really not unknown in view of increasing relevant literatures, but with deep uncertainty.

Line 24-25: "In some regions of … (warm cavity ice shelves)"_please add some supporting references.

Line 63: which "two time periods"?

Line 141-142: "two scenarios: i) … (AFTER experiment)."_confusing statement: you mean the periods of January 2010 to July 2016 and July 2016 to December 2020 respectively for the BEFORE and AFTER experiments?

Line 179: "where the surface temperature ($T0$) is based on the surface salinity ($S0$)"_you should explicitly state that the surface temperature is set to the surface freezing point.

Line 186-188: "The salinity is in LADDIE, however, described by a quadratic function, … as for the temperature, following"_It appears to be some redundant to mention the quadratic function.

Line 197: "Ocean models approximate physical processes, which implies that they

need to be tuned in order to match observations."_the logical relationship between the former and latter clauses is confusing.

Line 206: "The basal melt rate trend from 2010 to 2022, shown in Fig. 1,"_actually, no trend is shown in Fig. 1.

Line 230-231: "in both 2014/15 and 2015/16 (Fig. 5a and b) from ~370 m/yr to almost ~400 m/yr"_such a large increase in ice speed only occurred in 2014/15; "almost ~400 m/yr" => "395 m/yr".

Line 241: "revealing a regime shift from cold and fresh conditions to warmer and saltier conditions across all profiles" => "revealing both warming and salinization trends across all profiles", and you should indicate the depth range for that; the surface layer is the exception.

Line 304: ", like Pine Island and Totten, which both have a higher projected potential sea level rise contribution."_ please add some supporting references.

**Typos:**

Line 25: "warm cavity ice shelves" => "warm-cavity ice shelves" throughout the whole text

Line 53: give the full spelling of "REMA" for the first appearance

Line 126: "has a monthly temporal resolution" => "has a monthly temporal output"?

Line 176: "Austral winter months" => "austral winter months"

Line 252: "have their outflow" => "have their outflows"

Line 298-299: "in-situ field measurements" => "in-situ measurements" or "field measurements"

**Figures:**

Fig. 1: in caption "the old and new channel" => "the old and new channels"; "by the white square shows" => "by the white square show"; "Two similar zoom-ins of the highly channelized area marked by the white square"_these two zoom-ins are the same, aren't they?

Fig. 2: I cannot fully understand the correlation between the statement and the corresponding diagram in each subplot, which needs to be much more understandable and clearer.

Fig. 7: in caption "which roughly corresponds to the layers" => "which roughly correspond to the layers"

Fig. 8: in caption the arrows are also shown in (b) and (e).

**References:**

Cheng, C., Jenkins, A., Holland, P. R., Wang, Z., Dong, J., & Liu, C. (2024). Ice shelf basal channel shape determines channelized ice-ocean interactions. Nature Communications, 15, 2877.

Gladish, C. V., Holland, D. M., Holland, P. R., & Price, S. F. (2012). Ice-shelf basal channels in a coupled ice/ocean model. Journal of Glaciology, 58, 1227-1244.

Gourmelen, N., Goldberg, D. N., Snow, K., Henley, S. F., Bingham, R. G., Kimura, S., ... & van de Berg, W. J. (2017). Channelized melting drives thinning under a rapidly melting Antarctic ice shelf. Geophysical Research Letters, 44, 9796-9804.

Lambert, E., Jüling, A., van de Wal, R. S. W., & Holland, P. R. (2023). Modelling Antarctic ice shelf basal melt patterns using the one-layer Antarctic model for dynamical downscaling of ice-ocean exchanges (LADDIE v1.0). The Cryosphere, 17, 3203-3228.

Millgate, T., Holland, P. R., Jenkins, A., & Johnson, H. L. (2013). The effect of basal channels on oceanic ice-shelf melting. Journal of Geophysical Research: Oceans, 118, 6951-6964.

Payne, A. J., Holland, P. R., Shepherd, A. P., Rutt, I. C., Jenkins, A., & Joughin, I. (2007). Numerical modeling of ocean-ice interactions under Pine Island Bay's ice shelf. Journal of Geophysical Research: Oceans, 112, C10019.

Sergienko, O. V. (2013). Basal channels on ice shelves. Journal of Geophysical Research: Earth Surface, 118, 1342-1355.

---

## Referee Comment (RC2)

**Review of egusphere-2025-573**

*Overview:*

The manuscript "Ocean-Induced Weakening of George VI Ice Shelf" by Zinck et al. describes the formation of a new channelized surface feature on the George VI Ice Shelf. The authors estimate basal melting rates from remote sensing observations and compare these to modelled melt rates. Ocean model output (temperature and salinity) is presented to uncover possible drivers of enhanced channelization. They also investigate whether the feature could involve fracture propagation by examining time series of ice flow divergence, although I (and the authors) am left unconvinced one way or the other. The paper is well-written, focused, and not too long. I find the observations of how this complex channelized system is evolving to be intriguing, timely, and valuable information for the community. I have several specific comments to consider below.

*Specific comments:*

1. Line 8: "channel re-routing... with the channel serving as a basal melt channel" Do you mean to specify that the *new* channel is serving as a basal melt channel?
2. Line 111: You are using a different velocity product than in the BURGEE calculations. Is this line the reason for this choice? Why not use the same velocity product throughout? Some clarification would be good.
3. Line 142: I know this might seem obvious, but I was confused at first why you named the experiments BEFORE and AFTER. Before and after what, exactly? The emergence of the new channel? Please clarify.
4. Line 140: Here you should briefly describe the physics/assumptions/equations that the LADDIE model is based on.
5. Line 152: This statement about non-ice-shelf areas is unclear. I already know you are only looking at the ice shelf so maybe just remove this.
6. Figure 2: This figure did not help me understand the workflow any more than the basic description in the text. The sequence of different shapes and arrows did not make sense to me. If you could make a similar figure about the workflow with actual data, that would be more insightful.
7. Figure 3b: Are the colors for BEFORE and AFTER incorrect here? For MITgcm, red (2020) has higher salinity than blue (2010). But AFTER (red) has lower salinity than BEFORE (blue) here, which is especially confusing given the time series in Figure 6.
8. Figure 3 caption: Change "temperate" to "temperature"

9. Section 3.4.3: I'm wondering what a typical range of values is for the drag coefficient and how the value found from tuning to BURGEE fits within this range.

10. Table 2: Specify "Ice temperature -25 C" is referring to ice surface temperature? Where did you get this value from?

11. Line 223: "flanking uplift is typically associated with fracture"... Actually, this type of "flanking uplift" can arise for narrower channels (relative to ice thickness) without any fracture or extensional stresses, in a purely viscous model (see Stubblefield et al., 2023). So flanking uplift on its own does not imply fracture or extension.

12. Related to previous comment: I'm wondering if there is any surface imagery that shows fracture patterns in this area.

13. Divergence: Is this referring to div(thickness*velocity) or thickness*div(velocity)? div(velocity) on its own (as described in section 3.2) should have units of 1/yr, right? Here, the divergence units are always reported as m/yr though.

14. Figure 4: I'd like to see the surface elevation profiles along an additional transect (like panel j) at the other side of the new channel (i.e. left side in image). I'm curious if the rate of elevation change along this new channel is mostly uniform or not. From the color maps, it looks like it emerges uniformly along its length over time, but it is hard to tell for certain. This could provide some clues about the more detailed physics. Suggest also adding analogous panels to Figure 5.

15. Line 228: It's important to note, at least in the discussion, that channels themselves can generate viscous flow independent of any fracturing (Wearing et al., 2020). The divergences you are reporting could originate from viscous flow generated by channelization, especially since they are small in magnitude.

16. Line 248: I think these statements about ENSO should be left for the discussion because it is not a result of this study. Unless you want to also show an ENSO index and include a timeseries analysis or something to further support this idea.

17. Line 253: "possibly indicating increased meltwater outflow". I was confused whether the MITgcm ocean model is being forced by glacial meltwater inputs? If so, it seemed like this could be tracked down. However, I was a bit confused what this could demonstrate about temporal evolution of channels anyways because you said that MITgcm has a fixed ice geometry. Some clarification is necessary here.

18. Figure 7: Should specify that these results are from MITgcm. Also, the yellow trace of the channel in panel c seems to be between the positive and negative areas, while in panel f it is in the negative area. The differences for the different depths are not described in the main text, where you just say "higher current velocities near the channel", but it seems like it might be more complex than that.

19. Line 267: You claim a "strong agreement" between the modelled melt rate and observations (at least in part because you tuned the model parameter). I wanted to

see a direct comparison between the BURGEE and LADDIE melt rates (e.g., plot side-by-side and/or subtract colormaps), and some more quantitative metrics. The maximums should be close because those were used for tuning, but what about the mean or the variability, etc.?

20. Are the ocean velocities in Figure 7 and Figure 8 different types of velocities? I wasn't sure exactly what plume velocity means, for example. I'm just wondering if a direct comparison between the flow fields makes any sense or not.

21. Figure 8: Specify that these melt rates are from LADDIE (as opposed to BURGEE).

22. Discussion: I think the discussion about possible ENSO relations needs more detail. I was looking at the Boxall et al. (2024) paper and I think the many *Cryosphere* readers would benefit from more background on this and how it relates to your observations.

23. Line 282: The phrase "both the latter" is unclear to me.

24. Line 283: As previously stated, the uplift and divergence variations are not necessarily exclusive to fracturing; they can arise in a purely viscous secondary flow induced by channelization. I am not convinced that the ice-flow or divergence timeseries point to fracturing, but I still think it is valuable information to include in Figure 5.

25. An interesting point of this study is the emergence of a new channel in a highly channelized area. It even cuts across (or emerges from) a preexisting channel. I think the interaction between multiple channels would be an interesting topic to ponder or discuss further. I'm wondering if the preexisting channels set up a preferential flow pathway for the plume to carve out a new channel. I'm also interested in how the stresses in the ice from new channel interact with the preexisting channel in terms of the "structural integrity" of the ice shelf (thinking of Figure 4 in Drews 2015).

**References:**

- Stubblefield, A. G., Wearing, M. G., & Meyer, C. R. (2023). Linear analysis of ice-shelf topography response to basal melting and freezing. *Proceedings of the Royal Society A*, *479*(2277), 20230290.

- Wearing, M. G., Stevens, L. A., Dutrieux, P., & Kingslake, J. (2021). Ice-shelf basal melt channels stabilized by secondary flow. Geophysical Research Letters, 48(21), e2021GL094872.

- Drews, R. (2015). Evolution of ice-shelf channels in Antarctic ice shelves. *The Cryosphere*, *9*(3), 1169-1181.

---

## Author Response (AR1)

**Dear Jan de Rydt,**

Thank you very much for the constructive feedback, which helped a lot in revising the manuscript. Below we have outlined the changes we have made to the manuscript based on your suggestions.

**1. Timeseries:**

In the revised manuscript we have removed the ocean circulation figure, as it did not add much to the story, and because of the clear scale/resolution difference between the model and the channel. Furthermore, we have added a more nuanced discussion on how these models used cannot definitively demonstrate the causality of the channel-emergence. As suggested, we have also added a timeseries which compares the Oceanic Nino Index with MITgcm basal melt rates and BURGEE channel deepening rates.

**2. Spatial maps:**

As stated above to point 1 on timeseries, the MITgcm circulation figure has been removed from the manuscript and a new timeseries figure has been added to the manuscript.

We have done as suggested with regards to spatial maps of effective strain rates and added yearly effective strain rate maps to the manuscript. These maps show the same picture as the cross section investigation of divergence across the channel.

**3. Disparity of scales:**

The newly added timeseries includes temporal changes in MITgcm modelled basal melting averaged over the entire ice shelf and of the channel-are, respectively, thus avoiding the disparity of scales as was one of the problems with the ocean circulation figure. In the revised manuscript we have made sure to also acknowledge the scale issues of both LADDIE (steep slopes) and BURGEE (hydrostatic equilibrium).

Based on your comments and the comments from the reviewers the main changes to the manuscript thus are:

- A new timeseries comparing Oceanic Nino Index, MITgcm basal melt rates and BURGEE channel deepening rates including a broader discussion on El Nino.
- A new spatial map of effective strain rates.
- LADDIE sensitivity tests.
- A clearer acknowledgement of the disparity of scales.
- A new movie of optical imagery of the channel-area.

On behalf of all authors, Ann-Sofie P. Zinck

**Reviewer 1**

Review of "Ocean-Induced Weakening of George VI Ice Shelf" by Zinck et al.

**General comments:**

In this study, the authors uncovered a newborn basal channel within the extensive channelized network of the southern frontal area of George VI Ice Shelf. That channel exhibits remarkably higher basal melt rate than the surrounding area, and may have the potential to influence the ice shelf weakening. The authors argue that the emergence of that channel is likely linked to the 2015 ENSO event. I think the manuscript is well written and structured, and the uncovering of the new channel is intriguing, but I also do think the mechanism behind the generation of that channel and its influence on the ice-shelf instability are poorly demonstrated by the oceanic models the authors used in view of the quite inadequate horizontal resolution, fixed thermohaline forcing, inconsistence between the model parameter and the modeled results, and the insignificant thermal and speed differences between the BEFORE and AFTER experiments, etc. Please refer to my major comments for more details. As such, I regret that I could not support to publish this manuscript in its current form in The Cryosphere. Some minor suggestions are also provided.

**Thank you for the review.**

In this manuscript we have discovered a new channel on George VI through high-resolution remote sensing surface elevations. These high-resolution elevations are further used to calculate the basal melt rates using BURGEE (Zinck et al., 2023, 2024). The further purpose of this manuscript is then to investigate the potential drivers behind this fast-developing channel, with the hypothesis that the channel must either be a fracture, a basal melt channel, or a combination of the two, by making use of existing data and modelling results. To investigate the fracture theory we utilize available surface velocity data to explore temporal changes in speed and divergence across the channel. Secondly, to investigate changes in ocean heat and circulation within the ice shelf cavity we use MITgcm-based model results which allow us to investigate the changes in available (modified) circumpolar deep water, and thus the changes in ocean heat available to drive basal melting. Lastly, we utilize LADDIE to investigate potential changes in plume direction and melt pattern to be able to assess whether or not a melt plume is likely to follow the new channel, given its eccentric geometry and connection to the existing channel network.

We are, therefore, of the belief that there was a misunderstanding/misalignment between the goal of the manuscript and the reading of the reviewer. We have outlined that in the response to each of the major comments, where we also address how we aim to clarify this in the updated manuscript if applicable.

All changes made to the revised manuscript are marked in orange. Comments marked with just "Completed" mean that we have revised the manuscript exactly according to our original comment in blue.

**Major comments:**

First of all, the adopted horizontal spacing in LADDIE is 500 m, and that in the previous MITgcm-based simulation the authors utilized is much coarser, that is, 2-4 km. In contrast, the width of the new channel unveiled here is rather narrow, that is, only 2 km (as shown in Fig. 4j). That means such a fine basal topography is not represented at all in the utilized MITgcm model, and is also poorly represented in the LADDIE. Therefore, those coarse horizontal resolutions are far from adequate to resolve the interaction between the new channel and the underlying ocean.

It is absolutely correct that the resolution of the MITgcm-based simulation results which we use is too coarse to capture the new channel, as well as most other channels on GeorgeVI which are seldom wider than a few kilometers. That is why we only use the results for these two purposes:

- i) To explore changes in ocean heat and salinity at each of the entrances to the ice shelf cavity as well as in the near vicinity of the new channel. Temperature and salinity anomalies from all three transects (Fig. 6) show a similar picture of, e.g., increasing temperature, throughout the study period. A higher spatial resolution will not change this conclusion, and we also do not draw any conclusions with regards to the temperature and salinity which are based on the spatial resolution of the model.
- ii) To explore changes in ocean circulation in the vicinity of the new channel. In L254-258 we write

"In addition to changes in ocean temperature and salinity, MITgcm model outputs suggest alterations in ocean circulation near the channel (Fig. 7). Although there is a disparity in scale between the model grid size and the size of the channel, we can reasonably conclude that changes in ocean circulation likely occurred near the channel between 2010 and 2020, with higher current velocities near the channel. However, interpreting circulation changes in more detail and on a smaller scale, particularly in the immediate vicinity of the channel, remains challenging given the coarse model resolution."

We make it very clear that there is a disparity in scale between the model resolution and the size of the channel, and that all we can reasonably conclude is that changes **likely** did occur in the ocean circulation near the channel, but that we cannot draw any conclusions in the near vicinity of the channel due to the differences in resolution.

In the revised manuscript we will make sure to clarify this disparity in scale already in Sect 3.3. We will likewise emphasize our goal of using the MITgcm model results, which will help clarify this further.

We agree with the reviewer that a higher horizontal resolution in LADDIE would allow for better-resolved flow dynamics and a more accurate estimate of plume velocity, and consequently melt rates. However, since the primary focus of our study is to assess whether the meltwater flow is directed through the channel — rather than to precisely quantify its speed or consequent melt rate — we believe that a resolution of 500 m is sufficient for capturing the key pathway of the flow. We will clarify this in Sect. 3.4 of the revised manuscript.

In the revised manuscript we acknowledge this limitation further:

"The new channel is narrow, whereas LADDIE has been primarily validated in broader channels with gently sloping sidewalls (Lambert et al., 2023). Flow in narrow, steep-sided channels may involve additional complexity, including ageostrophic processes (Millgate et al., 2013), that may not be fully captured in our setup. While we consider our current configuration sufficient for assessing plume pathways, caution is warranted when interpreting the exact melt rates and plume speeds in the channel."

Second, as shown in Fig. 6b and h and Fig. 7c and f, the thermal and speed differences just below the ice base of the focused area between the BEFORE and AFTER periods are no more than 0.2 oC and 2 cm/s, respectively. Those small anomalies I think would not account for "the rapid emergence of a significant channel" with sufficient credibility.

The temperature anomaly in Fig 6b just below the ice base ranges from -0.28 to 0.27 degree C (so a difference of 0.55 degree C) and the speed difference of the area plotted in Fig. 7 ranges from -2.6 cm/s to 3.2 cm/s (a difference of 5.8 cm/s) in Fig. 7c and from -2.0 cm/s to 2.1 cm/s (a difference of 4.1 cm/s) in 7f. Whether or not these values are sufficient to cause the rapid emergence of the channel is not something which we claim in the manuscript. We use the temperature anomaly as a proxy of the availability of (modified) circumpolar deep water, and draw the conclusion that the entire water column within the ice shelf cavity has warmed within our study period, and that this regime shift, as well as the emergence of the channel, coincide with the 2015 ENSO event. In the revised manuscript we will ensure to emphasize this latter point.

In the revised manuscript we have, furthermore, included a time series of MITgcm melt rates, channel deepening rates and oceanic nino index.

Third, the adoption of the constant (only winter-averaged in 2010 and 2020) profiles to force LADDIE seems to be some inappropriate in view of the fast-evolving southern part of George VI Ice Shelf as illustrated in this study.

We assume that the reviewer is referring to the fact that we use the winter-average of the two years (2010 and 2020) as opposed to the entire year. As mentioned in the text in L176-177 "We only consider Austral winter months (May-August) to reduce the noise level from the seasonality in the upper ocean layers." Furthermore, melting is typically initiated at the grounding line, and it is thus the temperature at the depth of the grounding line which is the more important factor when initializing melt plumes in LADDIE. In the Figure below which is similar to Fig. 3 in the manuscript, we have added a dotted line which indicates the average MITgcm temperature/salinity when considering the whole year as opposed to only the summer months (dashed lines). From this figure it can be seen that only the uppermost layers (0 to -100 m depth) are impacted by our choice of using winter months only.

In the updated manuscript we will emphasize that LADDIE is designed to simulate steady-state melt patterns for a given geometry and ambient ocean forcing and that we therefore want to compare the before and after state in a 'snapshot-manner'. We thus give two fixed profiles which are representative for the BEFORE phase and the AFTER phase, in between which we see a clear switch from cold to warm temperatures (Fig 6). Completed

Fourth, there is a significant inconsistence between the large channelized basal melt rate (up to 30 m s -1 as shown in Fig. 1) and both low drag coefficient (310 -4) and plume speed (the maximum is only 0.15 m/s as shown in Fig. 8b and e), which would make me question the simulated results of LADDIE. At last, the LADDIE results shown in Fig. 8, for me, do not provide any new insights into the channelized meltwater plume and the corresponding basal melting. The plume tends to be topographically guided by the basal channels, and the basal melting is amplified therein. That finding has been proved in a wealth of literatures as listed at the end of this report.

We used an extend-triangle in the colorbar to indicate that higher and/or lower values are also present. We will clarify this better in the revised version and mention it explicitly in the figure captions where applicable. This implies that the maximum plume velocity is not 0.15 m/s as the reviewer suggests. The highest value within the new channel is 0.25 m/s, which together with the used drag coefficient and local plume temperature correspond to a melt rate above 30 m/yr when using the three-equation formulation (Holland and Jenkins, 1999)

which is adopted in LADDIE (Lambert et al., 2023) but also by other similar models (e.g. Sergienko, 2013). In the updated manuscript we will extend the colorbar in Fig. 8b and 8e to 0.25 m/s to allow for better visualisation of the plume velocity.

To avoid confusion, we will make sure that all colorbars get adjusted throughout the manuscript, to ensure that the colorbars are only closed if no values are exceeding it, and otherwise with arrows as they are in Fig. 8. This change will apply to Fig. 6 and 7. We will likewise stress this in the caption.

It is correct that plumes tend to be topographically guided and that this finding is not new. However, the new channel is of a very obscure geometry, cutting through the old channel. It is therefore not a given that the meltwater plume would use it as a pathway. Secondly, the point of this manuscript is to investigate all possible origins and causes of this channel, and not including LADDIE would therefore give an incomplete picture. Further, the LADDIE results explain and support the remote-sensing-derived basal melt pattern which shows a combination of the BEFORE and AFTER LADDIE melt patterns, as mentioned in L264-268. We have taken care of the colorbars in all figures. Furthermore, we have added a sensitivity experiment of the drag coefficient to the newly added Appendix.

Therefore, after reviewing this manuscript I still have no idea why that channel was born, and what it will bring about for the upcoming evolution of George VI Ice Shelf. I think that could not be simply attributed to ocean warming, the link of which to ENSO, however, remains unclear (as stated in Line 277). Although the authors have acknowledged that "there is a disparity in scale between the model grid size and the size of the channel (Line 255)" and "interpreting circulation changes in more detail and on a smaller scale, particularly in the immediate vicinity of the channel, remains challenging given the coarse model resolution (Line 257-258)", it is the model flaws they mentioned above that are **SO** critical to resolve my concern. In that sense, maybe the coupled ice sheet-ocean models (as used in Gladish et al., 2012 and Sergienko, 2013) but with sufficiently high resolution would be an effective way.

As mentioned previously, the goal of the manuscript is to investigate all of the possible drivers behind the rapid channel-formation as well as shedding light on its nature (fracture, melt channel, or a combination). To confidently be able to answer why the channel was born most likely does require high-resolution coupled ice-ocean modelling. However, since we do not know whether this channel is a fracture, a basal melt channel or a combination of both, it would require an ice sheet model with a sophisticated fracturing mechanism scheme, a detailed ocean model and a solid coupling of the two in a realistic setup and not just an idealized one. Such a model does not yet exist - at least not in a published version - and that is also out of the scope of this manuscript.

We do not attribute the sole reason for its emergence to ocean warming and ENSO. We conclude that its appearance coincides with changes in both ocean forcing and ice divergence and that both of these align with ENSO: "The appearance of the channel aligns with both changes in ocean forcing, most notably increased ocean temperatures and salinity, and subtle changes in ice divergence, both of which coincide with the timing of a major ENSO event. While the exact link between ENSO and the development of this channel remains speculative, the temporal correlation suggests that large-scale climate patterns may have a role in amplifying basal melting and possibly in re-routing meltwater pathways on

Antarctic ice shelves." (L307-311). We will stress this better in the revised version. The revised manuscript includes a timeseries which compares the oceanic nino index with MITgcm melt rates and the REMA channel deepening rate.

**Specific comments:**

The title: I think the temporal scale of this weakening should be specified.

We could potentially change the title to "Ocean-Induced Weakening of George VI Ice Shelf between 2010 and 2022". However, we are afraid that including a specific temporal scale in the title could be interpreted as if we suggest that the weakening has stopped. We will leave it to the editor to decide on which title is more appropriate.

We have not made any changes to the title of the manuscript.

Line 16: "..., largely due to the unknown response of ice shelves."\_response to what? From my point of view, the response of ice shelves to the changing oceanic and climatic conditions is really not unknown in view of increasing relevant literatures, but with deep uncertainty. We will change it to "..., largely due to the uncertain response of ice shelves." Completed

Line 24-25: "In some regions of ... (warm cavity ice shelves)"\_please add some supporting references.

We will add the following reference: Silvano et al., 2016 Completed

Line 63: which "two time periods"?

We will change it to "two time periods (2010-2016 and 2016-2022)". Completed

Line 141-142: "two scenarios: i) ... (AFTER experiment)."\_confusing statement: you mean the periods of January 2010 to July 2016 and July 2016 to December 2020 respectively for the BEFORE and AFTER experiments?

We will change it to "two scenarios: i) January 2010 to July 2016 (BEFORE experiment), and ii) July 2016 to December 2022 (AFTER experiment)."

We have rephrased this to: "... two scenarios corresponding to before and after the new channel emerged: i) January 2010 to July 2016 (pre-emergence experiment), and ii) July 2016 to December 2022 (post-emergence experiment)."

Line 179: "where the surface temperature (T0) is based on the surface salinity (S0)"\_you should explicitly state that the surface temperature is set to the surface freezing point. We will change it to "the surface temperature (T0) is set to the surface freezing point based on the surface salinity (S0)..."

Completed

Line 186-188: "The salinity is in LADDIE, however, described by a quadratic function, ... as for the temperature, following"\_It appears to be some redundant to mention the quadratic function.

We will exclude the part on the quadratic function and instead write "We use a similar tangent hyperbolic function for the salinity (S), following..."

Completed

Line 197: "Ocean models approximate physical processes, which implies that they need to be tuned in order to match observations."\_the logical relationship between the former and latter clauses is confusing.

We will change it to "Because ocean models approximate physical processes, they require tuning to better match observations."

Completed

Line 206: "The basal melt rate trend from 2010 to 2022, shown in Fig. 1,"\_actually, no trend is shown in Fig. 1.

We will change it to "The basal melt rate from 2010 to 2022, shown in Fig. 1". Completed

Line 230-231: "in both 2014/15 and 2015/16 (Fig. 5a and b) from ~370 m/yr to almost ~400 m/yr"\_such a large increase in ice speed only occurred in 2014/15; "almost ~400 m/yr" => "395 m/yr".

We will change it to "Our analysis of ice speed and divergence across the channel show speeds along the transect fluctuating substantially in both 2014/15 (372 m/yr to 395 m/yr, Fig. 5a) and 2015/16 (375 m/yr to 397 m/yr, Fig. 5b) with an isolated peak in the ice speed at the channel location in 2015/16."

Completed

Line 241: "revealing a regime shift from cold and fresh conditions to warmer and saltier conditions across all profiles" => "revealing both warming and salinization trends across all profiles", and you should indicate the depth range for that; the surface layer is the exception. We will change it to "revealing a regime shift from cold and fresh conditions to warmer and saltier conditions below the surface layer across all profiles." We have chosen to keep the original structure of this sentence to emphasize the regime shift.

Completed

Line 304: ", like Pine Island and Totten, which both have a higher projected potential sea level rise contribution."\_ please add some supporting references.

We will add and refer to the following two references: Seroussi et al. 2020 and 2023.

Completed

**Typos:**

Line 25: "warm cavity ice shelves" => "warm-cavity ice shelves" throughout the whole text We will change it according to your suggestion.

Completed

Line 53: give the full spelling of "REMA" for the first appearance We will change it according to your suggestion. Completed Line 126: "has a monthly temporal resolution" => "has a monthly temporal output"? We will change it to "has a monthly temporal output".

Completed

Line 176: "Austral winter months" => "austral winter months" We will change it according to your suggestion.

Completed

Line 252: "have their outflow" => "have their outflows" We will change it according to your suggestion.

Completed

Line 298-299: "in-situ field measurements" => "in-situ measurements" or "field measurements"

We will change it to *"field measurements"*. Completed

**Figures:**

=>

Fig. 1: in caption "the old and new channel" => "the old and new channels"; "by the white square shows" => "by the white square show"; "Two similar zoom-ins of the highly channelized area marked by the white square"\_these two zoom-ins are the same, aren't they?

We will apply the following changes: "the old and new channel" => "the old and new channels"; "by the white square shows" => "by the white square show".

Yes, the two zoom-ins are indeed similar, but with different things marked on them. We found that adding all of the needed "labels" made it impossible to actually get a zoom-in of the melt rate in the channel, so that is the reason for the two similar zoom-ins, as also stated in the figure itself. We suggest to do the following changes in the caption:

"Two similar zoom-ins of the highly channelized area marked by the white square shows BURGEE melt rates both with the approximate location of the surface depression of the old and new channel in year 2016 in teal, the new channel pointed out by the white arrow, as well as the red transect used in Fig. 4 and 5."

"Two similar zoom-ins of the highly channelized area marked by the white square show BURGEE melt rates. The upper zoom-in further shows the approximate location of the surface depression of the old and new channels in year 2016 in teal as well as the red transect used in Fig. 4 and 5. The lower zoom-in allows for better visualisation of the melting within the channel with the new channel pointed out by the white arrow."

Completed

Fig. 2: I cannot fully understand the correlation between the statement and the corresponding diagram in each subplot, which needs to be much more understandable and clearer.

In the updated manuscript we will adjust Sect. 3.4.1 such that all steps presented in the text are linked to a specific sub-panel of Fig. 2. Example of how we will do that L155-158:

"For the surface elevation for the BEFORE geometry, all strips up to 2016-07-01 are firstly displaced to their location as of 2013-01-01 using MEaSURES ITS\_LIVE velocities (Fig. 2a, Gardner et al., 2022). Secondly, these strips are further displaced using feature tracking relative to the median elevation map between 2012-07-01 and 2015-07-01 to ensure alignment across strips (Fig. 2b)."

We have eventually chosen to discard this figure and instead rewritten the section describing the creation of ice geometry heavily to improve the readability. We found that including a figure caused more confusion than help.

Fig. 7: in caption "which roughly corresponds to the layers" => "which roughly correspond to the layers"

We will change it according to your suggestion.

We have chosen to remove this figure from the revised manuscript as it did not add much to the story. Instead we have included a timeseries of MITgcm basal melt rates, REMA elevation change and Oceanic Nino Index.

Fig. 8: in caption the arrows are also shown in (b) and (e). We will adjust the caption accordingly.

Completed

**References:**

Cheng, C., Jenkins, A., Holland, P. R., Wang, Z., Dong, J., & Liu, C. (2024). Ice shelf basal channel shape determines channelized ice-ocean interactions. Nature Communications, 15, 2877.

Gladish, C. V., Holland, D. M., Holland, P. R., & Price, S. F. (2012). Ice-shelf basal channels in a coupled ice/ocean model. Journal of Glaciology, 58, 1227-1244.

Gourmelen, N., Goldberg, D. N., Snow, K., Henley, S. F., Bingham, R. G., Kimura, S., ... & van de Berg, W. J. (2017). Channelized melting drives thinning under a rapidly melting Antarctic ice shelf. Geophysical Research Letters, 44, 9796-9804.

Holland, D. M., & Jenkins, A. (1999). Modeling Thermodynamic Ice—Ocean Interactions at the Base of an Ice Shelf. Journal of Physical Oceanography, 29(8), 1787-1800.

Lambert, E., Jüling, A., van de Wal, R. S. W., & Holland, P. R. (2023). Modelling Antarctic ice shelf basal melt patterns using the one-layer Antarctic model for dynamical downscaling of ice-ocean exchanges (LADDIE v1.0). The Cryosphere, 17, 3203-3228.

Millgate, T., Holland, P. R., Jenkins, A., & Johnson, H. L. (2013). The effect of basal channels on oceanic ice-shelf melting. Journal of Geophysical Research: Oceans, 118, 6951-6964.

Payne, A. J., Holland, P. R., Shepherd, A. P., Rutt, I. C., Jenkins, A., & Joughin, I. (2007). Numerical modeling of ocean-ice interactions under Pine Island Bay's ice shelf. Journal of Geophysical Research: Oceans, 112, C10019.

Sergienko, O. V. (2013). Basal channels on ice shelves. Journal of Geophysical Research: Earth Surface, 118, 1342-1355.

Seroussi, H., Nowicki, S., Payne, A. J., Goelzer, H., Lipscomb, W. H., Abe-Ouchi, A., Agosta, C., Albrecht, T., Asay-Davis, X., Barthel, A., Calov, R., Cullather, R., Dumas, C., Galton-Fenzi, B. K., Gladstone, R., Golledge, N. R., Gregory, J. M., Greve, R., Hattermann, T., Hoffman, M. J., Humbert, A., Huybrechts, P., Jourdain, N. C., Kleiner, T., Larour, E., Leguy, G. R., Lowry, D. P., Little, C. M., Morlighem, M., Pattyn, F., Pelle, T., Price, S. F., Quiquet, A., Reese, R., Schlegel, N.-J., Shepherd, A., Simon, E., Smith, R. S., Straneo, F., Sun, S., Trusel, L. D., Van Breedam, J., van de Wal, R. S. W., Winkelmann, R., Zhao, C., Zhang, T., and Zwinger, T.: ISMIP6 Antarctica: a multi-model ensemble of the Antarctic ice sheet evolution over the 21st century, The Cryosphere, 14, 3033–3070, https://doi.org/10.5194/tc-14-3033-2020, 2020.

Seroussi, H., Verjans, V., Nowicki, S., Payne, A. J., Goelzer, H., Lipscomb, W. H., Abe-Ouchi, A., Agosta, C., Albrecht, T., Asay-Davis, X., Barthel, A., Calov, R., Cullather, R., Dumas, C., Galton-Fenzi, B. K., Gladstone, R., Golledge, N. R., Gregory, J. M., Greve, R., Hattermann, T., Hoffman, M. J., Humbert, A., Huybrechts, P., Jourdain, N. C., Kleiner, T., Larour, E., Leguy, G. R., Lowry, D. P., Little, C. M., Morlighem, M., Pattyn, F., Pelle, T., Price, S. F., Quiquet, A., Reese, R., Schlegel, N.-J., Shepherd, A., Simon, E., Smith, R. S., Straneo, F., Sun, S., Trusel, L. D., Van Breedam, J., Van Katwyk, P., van de Wal, R. S. W., Winkelmann, R., Zhao, C., Zhang, T., and Zwinger, T.: Insights into the vulnerability of Antarctic glaciers from the ISMIP6 ice sheet model ensemble and associated uncertainty, The Cryosphere, 17, 5197–5217, https://doi.org/10.5194/tc-17-5197-2023, 2023.

Silvano, A., Rintoul, S., and Herraiz-Borreguero, L.: Ocean-Ice Shelf Interaction in East Antarctica, Oceanography, 29, 130–143, https://doi.org/10.5670/oceanog.2016.105, 2016.

Zinck, A.-S. P., Wouters, B., Lambert, E., and Lhermitte, S.: Unveiling spatial variability within the Dotson Melt Channel through high-resolution basal melt rates from the Reference Elevation Model of Antarctica, The Cryosphere, 17, 3785–3801, https://doi.org/10.5194/tc-17-3785-2023, 2023.

Zinck, A.-S., Lhermitte, S., Wearing, M., and Wouters, B.: Exposure to Underestimated Channelized Melt in Antarctic Ice Shelves, Nature Climate Change [in review], NCLIM-24071992, pp. 0–26, https://doi.org/10.21203/rs.3.rs-4806463/v1, 2024.

**Reviewer 2**

Review of egusphere-2025-573

Overview:

The manuscript "Ocean-Induced Weakening of George VI Ice Shelf" by Zinck et al. describes the formation of a new channelized surface feature on the George VI Ice Shelf. The authors estimate basal melting rates from remote sensing observations and compare these to modelled melt rates. Ocean model output (temperature and salinity) is presented to uncover possible drivers of enhanced channelization. They also investigate whether the feature could involve fracture propagation by examining time series of ice flow divergence, although I (and the authors) am left unconvinced one way or the other. The paper is well-written, focused, and not too long. I find the observations of how this complex channelized system is evolving to be intriguing, timely, and valuable information for the community. I have several specific comments to consider below.

We thank the reviewer for the positive and constructive review. Please see all our responses marked in blue below each of your comments.

All changes made to the revised manuscript are marked in orange. Comments marked with just "Completed" mean that we have revised the manuscript exactly according to our original comment in blue.

**Specific comments:**

- Line 8: "channel re-routing... with the channel serving as a basal melt channel" Do you mean to specify that the new channel is serving as a basal melt channel? Yes, we will change it to "channel re-routing ... with the new channel serving as a basal melt channel." Completed
- 2. Line 111: You are using a different velocity product than in the BURGEE calculations. Is this line the reason for this choice? Why not use the same velocity product throughout? Some clarification would be good.
  - For the ice divergence analysis across the channel, we use a higher-resolution velocity product based on SAR imagery (ENVEO), as it provides better spatial detail and higher temporal resolution, which is important for this localized, short-term investigation. In contrast, the BURGEE method requires a long-term mean velocity field representative of the full study period, for which the ITS\_LIVE product, based on optical imagery and spanning multiple years, is more appropriate. We will clarify this in the revised manuscript.

In the revised manuscript we have revised the sentence pointed out in Line 111 and added an extra sentence:

"Based on Sentinel-1 (synthetic aperture radar) imagery, these velocities offer higher spatial and temporal resolution, along with greater coverage, compared to velocity products based on optical imagery feature tracking such the MEaSURE ITS\\_LIVE velocities used in BURGEE. These Sentinel-1-based velocities are not used in BURGEE since they are only available from 2015 onwards and thus do not cover the

**entire BURGEE period."**

3. Line 142: I know this might seem obvious, but I was confused at first why you named the experiments BEFORE and AFTER. Before and after what, exactly? The emergence of the new channel? Please clarify.

Very good point. We will rename the two experiments to 'pre-emergence' and 'post-emergence' and clarify in the revised manuscript that it refers to the emergence of the channel.

We have rephrased this to: "... two scenarios corresponding to before and after the new channel emerged: i) January 2010 to July 2016 (pre-emergence experiment), and ii) July 2016 to December 2022 (post-emergence experiment)."

4. Line 140: Here you should briefly describe the physics/assumptions/equations that the LADDIE model is based on.

In the revised manuscript we will add the following to briefly describe the physics used in LADDIE:

"LADDIE solves the vertically integrated Navier-Stokes equations to compute the temperature, salinity, thickness, and horizontal velocities of the meltwater plume below the ice shelf. Basal melt rates are calculated using the three-equation formulation for melting and refreezing (Holland and Jenkins, 1999; Jenkins et al., 2010), which includes conservation of heat and salt, along with a constraint that keeps the ice—ocean interface at the local freezing point."

Completed

 Line 152: This statement about non-ice-shelf areas is unclear. I already know you are only looking at the ice shelf so maybe just remove this.
 We will do as suggested and remove this part about non-ice-shelf areas.
 Completed

6. Figure 2: This figure did not help me understand the workflow any more than the basic description in the text. The sequence of different shapes and arrows did not make sense to me. If you could make a similar figure about the workflow with actual data, that would be more insightful.

Thank you for the suggestion. We will make an updated figure with actual data where applicable and also refer directly to the different sub-panels in the text when they are mentioned. Example of how we will do that L155-158:

"For the surface elevation for the BEFORE geometry, all strips up to 2016-07-01 are firstly displaced to their location as of 2013-01-01 using MEaSURES ITS\_LIVE velocities (Fig. 2a; Gardner et al., 2022). Secondly, these strips are further displaced using feature tracking relative to the median elevation map between 2012-07-01 and 2015-07-01 to ensure alignment across strips (Fig. 2b)."

We have eventually chosen to discard this figure and instead rewritten the section describing the creation of ice geometry heavily to improve the readability. We found that including a figure caused more confusion than help.

7. Figure 3b: Are the colors for BEFORE and AFTER incorrect here? For MITgcm, red (2020) has higher salinity than blue (2010). But AFTER (red) has lower salinity than BEFORE (blue) here, which is especially confusing given the time series in Figure 6.

The colors are correct. When adjusting the tangent hyperbolic function to roughly match the MITgcm results it is impossible to get a perfect match. In the uppermost 50 m of the water column the red (AFTER) has much lower salinity than the blue (BEFORE) in both the tangent hyperbolic fit and the MITgcm results. From -50 m to roughly -200 m depth the MITgcm results fluctuate more than what can be captured by a tangent hyperbolic function, and we can therefore not accurately capture the part where the red MITgcm (AFTER) has a slightly higher salinity than the blue MITgcm (BEFORE). However, the salinity difference between the two has little impact as it is only in the uppermost 200 m of the water column, and not at grounding line depths where basal melting is initialised.

No further adjustments have been made to the manuscript based on this comment.

Figure 3 caption: Change "temperate" to "temperature"
 Will do.
 Completed

9. Section 3.4.3: I'm wondering what a typical range of values is for the drag coefficient and how the value found from tuning to BURGEE fits within this range.

The drag coefficient used in this study is lower than the typical range of 0.001–0.003 employed in ice sheet models (Jourdain et al., 2017; Mathiot et al., 2017; Rosevaer et al., 2022). A reduced value of Cdtop may give greater weight to plume temperature relative to plume velocity in determining melt rates. We adopted this lower value because it was necessary to match the integrated melt; however, we suspect this discrepancy arises from the direct extrapolation of ice shelf front temperature profiles into sub-shelf cavity conditions – where temperatures may in fact be colder and the thermocline deeper than currently assumed.

Despite this, we believe the low Cdtop value does not compromise our main conclusion from the LADDIE simulations: that the new ice shelf geometry enables basal meltwater flow through the newly formed fracture or channel.

We will make sure to acknowledge this in the revised manuscript. Nonetheless, we plan to do some sensitivity tests with LADDIE using forcing temperature profiles with a deeper thermocline and a Cdtop within the typical range, to verify that this does not significantly alter the results.

In the revised manuscript we have included a sensitivity test of the drag coefficient in the new appendix. The sensitivity test shows exactly what we hypothesized: that colder forcing allows for a higher drag coefficient, but that the lower drag coefficient does not change the conclusions which we draw from the LADDIE simulations.

10. Table 2: Specify "Ice temperature -25 C" is referring to ice surface temperature? Where did you get this value from?

The ice temperature in Tab. 2 is referring to the interior ice temperature, which we will clarify in the revised manuscript. The value of -25C has successfully been used in LADDIE to model melt rates under Crosson-Dotson and Filchner Ronne Ice Shelf (Lambert et al., 2023) and is in agreement with observations of the Filchner Ronne Ice Shelf (Rosier et al., 2018). We expect the ice temperature to have limited impact on the melt rates, but will perform a sensitivity study with higher ice temperatures which we will include in a Supplementary.

In the revised manuscript we have added a sensitivity test of the internal ice

temperature on the modelled basal melt rates to the new Appendix. The sensitivity test shows that the ice temperature only has a small impact on the magnitude of melting and does not impact the plume pathway.

11. Line 223: "flanking uplift is typically associated with fracture"... Actually, this type of "flanking uplift"can arise for narrower channels (relative to ice thickness) without any fracture or extensional stresses, in a purely viscous model (see Stubblefield et al., 2023). So flanking uplift on its own does not imply fracture or extension.

We thank the reviewer for pointing this out and for the useful reference! In the updated manuscript we will add a comment on how the flanking uplift is not guaranteed to be a result of fracturing as narrower channels can experience similar uplift.

The revised manuscript now includes this point:

"This type of bump, known as flanking uplift, is typically associated with fractures on ice shelves (Walker et al., 2019). However, narrower channels can cause similar flanking uplift (Stubblefield et al., 2023)."

12. Related to previous comment: I'm wondering if there is any surface imagery that shows fracture patterns in this area.

The new channel is visible in Sentinel 2 imagery and also from the optical imagery it remains a mystery as to whether it is fully a channel, fracture, or both. See example here (R.Fig. 1):

R.Fig. 1: Optical Sentinel 2 imagery from April 2022 of the study area.

In the revised manuscript we will include a timeseries of optical imagery from the channel area in the Supplement.

Completed

- 13. Divergence: Is this referring to div(thickness\*velocity) or thickness\*div(velocity)? div(velocity) on its own (as described in section 3.2) should have units of 1/yr, right? Here, the divergence units are always reported as m/yr though.
  - That is a typo from our side. The divergence (div(vel)) is calculated as described in section 3.2 and does indeed have the unit of 1/yr. We will make sure to fix those typos in the revised manuscript.

Completed

14. Figure 4: I'd like to see the surface elevation profiles along an additional transect (like panel j) at the other side of the new channel (i.e. left side in image). I'm curious if the rate of elevation change along this new channel is mostly uniform or not. From the color maps, it looks like it emerges uniformly along its length over time, but it is hard to tell for certain. This could provide some clues about the more detailed physics. Suggest also adding analogous panels to Figure 5.

R.Fig. 2 is an example of such a transect left of the original red transect as suggested (the new yellow transect is marked in panel e and below are the elevation profiles of that new yellow transect). The transect shows more or less the same pattern as the original red transect with the one difference that a part of the old channel has expanded, which results in the extensive surface lowering from the new channel and towards "B" in the transect. We, therefore, do not think that a second transect offers that much extra information and suggest keeping it out of the revised manuscript.

R.Fig. 2: Similar to Fig. 5 in the original manuscript, but here with elevation profiles of the yellow transect marked in e).

**No adjustment has been made to the manuscript based on this comment.**

- 15. Line 228: It's important to note, at least in the discussion, that channels themselves can generate viscous flow independent of any fracturing (Wearing et al., 2020). The divergences you are reporting could originate from viscous flow generated by channelization, especially since they are small in magnitude.
  In the Discussion at Line 283 we will add the following sentence: "Furthermore, it should be noted that channels themselves can generate viscous flow independent of any fracturing, which implies that both the observed flanking uplift as well as the subtle divergence signal could also purely be due to channelization."
  Completed
- 16. Line 248: I think these statements about ENSO should be left for the discussion because it is not a result of this study. Unless you want to also show an ENSO index and include a time series analysis or something to further support this idea. In the revised manuscript we will include an ENSO index and compare it to the MITgcm temperature and salinity timeseries as well as the ice shelf averaged basal melt rate computed by MITgcm.
  - The new time series compares the oceanic nino index with MITgcm mean and maximum basal melt rates across the entire ice shelf and of the channel area, and with channel deepening rates of the new channel.
- 17. Line 253: "possibly indicating increased meltwater outflow". I was confused whether the MITgcm ocean model is being forced by glacial meltwater inputs? If so, it seemed like this could be tracked down. However, I was a bit confused what this could demonstrate about temporal evolution of channels anyways because you said that MITgcm has a fixed ice geometry. Some clarification is necessary here. In the revised manuscript we will include the ice-shelf-wide averaged basal melt rate time series based on the MITgcm results, which directly shows the increased amount of basal melting from the ice shelf over time. This increase in meltwater, which causes freshening of the upper ocean layers as mentioned in Line 253 and shown in Fig. 6, does not directly demonstrate any temporal evolution of the channels. It shows that according to MITgcm the basal melt rate of the ice shelf has increased over the study period. As you correctly mention, the model uses a fixed geometry which implies that there is no change to channels. However, to create a new channel or to modify the pathway of an existing channel, increased melting is likely needed to force the meltwater plume to change its pathway. Or a fracture has to be present to serve as a favourable pathway for the plume. We will ensure to clarify this in the revised manuscript. MITgcm does get glacial meltwater input from the melting ice shelf and particle tracking experiments of meltwater-pathways have been conducted (Hyogo et al., 2024). However, because of the constant ice shelf geometry in the model, it makes little sense to track down a newly evolved and moving channel. Completed
- 18. Figure 7: Should specify that these results are from MITgcm. Also, the yellow trace of the channel in panel c seems to be between the positive and negative areas, while in panel f it is in the negative area. The differences for the different depths are not described in the main text, where you just say "higher current velocities near the

channel", but it seems like it might be more complex than that.

In the revised manuscript we will specify that these are MITgcm results. The reviewer is absolutely right that the circulation changes seem to be more complex than what we currently point out in the manuscript. In the updated manuscript we will, therefore, elaborate on the circulation changes at the two different depth levels and give a more nuanced picture.

We have chosen to exclude this figure from the revised manuscript, as it did not add much to the story. Instead we have included the timeseries of MITgcm basal melt rates, REMA elevation (changes), and Oceanic Nino Index as described in our updated response to point 16 and 17.

19. Line 267: You claim a "strong agreement" between the modelled melt rate and observations (at least in part because you tuned the model parameter). I wanted to see a direct comparison between the BURGEE and LADDIE melt rates (e.g., plot side by-side and/or subtract colormaps), and some more quantitative metrics. The maximums should be close because those were used for tuning, but what about the mean or the variability, etc.?

The LADDIE and BURGEE melt maps do not compare 1-1 in the geographical location of the channel due to the Lagrangian framework in BURGEE and the flow of the ice shelf. This also implies that subtracting the two from each other provides little information as the channel system will not be located in the same position. Likewise, the channel system is located in a different position in the two different LADDIE runs due to the flow of the ice in-between the two study periods. Therefore, it is also difficult to do a direct comparison of mean and variability as a geographically fixed study region will lead to including different parts of the channel system in BURGEE, LADDIE BEFORE, and LADDIE AFTER, respectively.

We have, however, run a "control" run of LADDIE where we use the BEFORE forcing on the AFTER geometry (B\_AFTER), and vice-versa the AFTER forcing on the BEFORE geometry (A\_BEFORE), which shows that the changes in melt pattern in LADDIE is controlled by geometry and not by forcing. As supplementary to the revised manuscript we will include this below figure (R.Fig. 3) which compares these control runs. In the supplementary we will likewise include a visual comparison of LADDIE and BURGEE melt rates.

R.Fig. 3: LADDIE forcing and geometry sensitivity..

Besides the control mentioned in our original reply we have also added a visual comparison of the LADDIE and BURGEE melt rates by adding a panel of BURGEE melt rates to the original LADDIE output figure. Furthermore, the control run has been added to the new Appendix.

20. Are the ocean velocities in Figure 7 and Figure 8 different types of velocities? I wasn't sure exactly what plume velocity means, for example. I'm just wondering if a direct comparison between the flow fields makes any sense or not.

MITgcm is 3D and models the ocean velocity at all depth layers in the model whereas, with a vertical resolution of 10 m near the surface to 450 m in the deepest layers. LADDIE, however, only models the velocity of the mixed layer below the ice base (plume velocity). The thickness of the mixed layer varies in each grid cell with a minimum thickness of 2 m in our simulations. These factors would have to be taken into account to make a somewhat direct comparison. However, the constant ice shelf geometry in MITgcm, which does not correspond to the two different geometries used in LADDIE, adds to the complexity of a direct comparison.

In the updated manuscript we will clarify what is meant by plume velocity, and how it differs from the ocean velocity in MITgcm.

In the revised manuscript we have chosen to remove Figure 7 as it did not add much to the story, and instead focus on the new timeseries.

- 21. Figure 8: Specify that these melt rates are from LADDIE (as opposed to BURGEE). Good point. We will do that. Completed
- 22. Discussion: I think the discussion about possible ENSO relations needs more detail. I was looking at the Boxall et al. (2024) paper and I think the many Cryosphere readers would benefit from more background on this and how it relates to your observations.

Thank you for the suggestion. We will make sure to make the ENSO discussion more detailed in the revised manuscript, and possibly also add a bit of background in the Introduction of the paper. As mentioned in point 16 above we will also add an ENSO index to the Results section which will also help address this issue further.

The revised manuscript includes a brief description of how ENSO impacts CDW in the Introduction and a more detailed and broader discussion on how ENSO impacts basal melting and Antarctica in general. Furthermore, the added timeseries (see point 16) has also contributed to a more detailed picture of the impact of ENSO in Antarctica.

23. Line 282: The phrase "both the latter" is unclear to me.

There was a typo in the sentence. We will rewrite the sentence to: "The observed new channel could represent a basal melt channel, a fracture, or a combination of both. Our investigations of the channel's origin point towards either a basal melt channel or a combination of both."

Completed

24. Line 283: As previously stated, the uplift and divergence variations are not necessarily exclusive to fracturing; they can arise in a purely viscous secondary flow induced by channelization. I am not convinced that the ice-flow or divergence timeseries point to fracturing, but I still think it is valuable information to include in Figure 5.

Please see our response to point 15 above. Further, we will revise the statement we made on line 283.

The revised manuscript reads:

"The observed new channel could represent a basal melt channel, a fracture, or a combination of both. Our investigations of the channel's origin point towards either a basal melt channel or a combination of both. The uplift in the surface near the depression, coupled with signs of stretching in the divergence field downstream of the channel in 2015/16, could suggest fracturing. However, these signals are subtle, leaving us unable to confirm or dismiss the possibility of fracturing with certainty. Furthermore, it should be noted that channels themselves can generate viscous flow independent of any fracturing (Wearing et al., 2020), which implies that both the observed flanking uplift as well as the subtle divergence signal could also purely be due to channelization."

25. An interesting point of this study is the emergence of a new channel in a highly channelized area. It even cuts across (or emerges from) a preexisting channel. I think the interaction between multiple channels would be an interesting topic to ponder or discuss further. I'm wondering if the preexisting channels set up a preferential flow

pathway for the plume to carve out a new channel. I'm also interested in how the stresses in the ice from new channel interact with the preexisting channel in terms of the "structural integrity" of the ice shelf (thinking of Figure 4 in Drews 2015).

Further studying the interaction of multiple channels would for sure be very interesting to dig deeper into. It would most likely require a rather sophisticated ice flow model which is able to both mimic fracturing and basal melting, and is thus out of the scope of this paper.

We have included a comment on this in the revised Discussion.

**References:**

Drews, R. (2015). Evolution of ice-shelf channels in Antarctic ice shelves. The Cryosphere, 9(3), 1169-1181.

Holland, D. M., & Jenkins, A. (1999). Modeling thermodynamic ice—ocean interactions at the base of an ice shelf. Journal of physical oceanography, 29(8), 1787-1800.

Hyogo, S., Nakayama, Y., & Mensah, V. (2024). Modeling ocean circulation and ice shelf melt in the Bellingshausen Sea. *Journal of Geophysical Research: Oceans*, 129, e2022JC019275. https://doi.org/10.1029/2022JC019275

Jenkins, A., Nicholls, K. W., & Corr, H. F. (2010). Observation and parameterization of ablation at the base of Ronne Ice Shelf, Antarctica. Journal of Physical Oceanography, 40(10), 2298-2312.

Jourdain, N. C., Mathiot, P., Merino, N., Durand, G., Le Sommer, J., Spence, P., ... & Madec, G. (2017). Ocean circulation and sea-ice thinning induced by melting ice shelves in the A mundsen S ea. Journal of Geophysical Research: Oceans, 122(3), 2550-2573.

Lambert, E., Jüling, A., van de Wal, R. S. W., and Holland, P. R.: Modelling Antarctic ice shelf basal melt patterns using the one-layer Antarctic model for dynamical downscaling of ice—ocean exchanges (LADDIE v1.0), The Cryosphere, 17, 3203–3228, https://doi.org/10.5194/tc-17-3203-2023, 2023.

Mathiot, P., Jenkins, A., Harris, C., & Madec, G. (2017). Explicit representation and parametrised impacts of under ice shelf seas in the z\* coordinate ocean model NEMO 3.6. Geoscientific Model Development, 10(7), 2849-2874.

Rosevear, M., Galton-Fenzi, B., & Stevens, C. (2022). Evaluation of basal melting parameterisations using in situ ocean and melting observations from the Amery Ice Shelf, East Antarctica. Ocean Science, 18(4), 1109-1130.

Rosier, S. H. R., Hofstede, C., Brisbourne, A. M., Hattermann, T., Nicholls, K. W., Davis, P. E. D., et al. (2018). A new bathymetry for the southeastern Filchner-Ronne Ice Shelf: Implications for modern oceanographic processes and glacial history. *Journal of Geophysical Research: Oceans*, 123, 4610–4623. https://doi.org/10.1029/2018JC013982

Stubblefield, A. G., Wearing, M. G., & Meyer, C. R. (2023). Linear analysis of ice-shelf topography response to basal melting and freezing. Proceedings of the Royal Society A, 479(2277), 20230290.

Wearing, M. G., Stevens, L. A., Dutrieux, P., & Kingslake, J. (2021). Ice-shelf basal melt channels stabilized by secondary flow. Geophysical Research Letters, 48(21), e2021GL094872.

---

## Referee Report (RR1)

2nd Review of "Ocean-Induced Weakening of George VI Ice Shelf" by Zinck et al.

**General comments:**

Thanks a lot for your revisions that satisfactorily address my major comments. At present, the results and discussions shown in this study are made much more robust and convincing than the previous ones after the authors carefully considered the comments from the reviewers and the editor. Thus, I think this manuscript is ready for publication in *The Cryosphere* just after a few minor revisions listed below.

**Specific comments:**

Title: "Ocean-Induced Weakening of George VI Ice Shelf" => "..., West Antarctica" or "... in West Antarctica"

The insertion of all the figures and tables should be placed after their first mentioning in the main text.

L. 203: "Because ocean models approximate physical processes, they require tuning to better match observations." The causality is still confused, just remove this sentence.

L. 241: "in both 2014/15 (372 m/yr to 395 m/yr, Fig. 4a) and 2015/16 (375 m/yr to 397 m/yr, Fig. 4b)" => "in both 2014/15 (minimum: 372 m yr-1, maximum: 395 m yr-1; Fig. 4a) and 2015/16 (minimum: 375 m yr-1, maximum: 397 m yr-1; Fig. 4b)"

Line 247: "the spatial analysis effective strain rates" => "the spatial analysis of effective strain rates"

Fig. 5: the channels marked in black are indiscernible from the base maps.

Fig. 6: what do the dashed lines in (h) and (i) denote?

---

## Author Response (AR2)

2nd Review of "Ocean-Induced Weakening of George VI Ice Shelf" by Zinck et al.

**General comments:**

Thanks a lot for your revisions that satisfactorily address my major comments. At present, the results and discussions shown in this study are made much more robust and convincing than the previous ones after the authors carefully considered the comments from the reviewers and the editor. Thus, I think this manuscript is ready for publication in The Cryosphere just after a few minor revisions listed below.

Thank you very much. Below we have pointed out all changes made below each comment.

**Specific comments:**

Title: "Ocean-Induced Weakening of George VI Ice Shelf" => "..., West Antarctica" or "... in West Antarctica"

We have changed the title to "Ocean-Induced Weakening of George VI Ice Shelf, West Antarctica".

The insertion of all the figures and tables should be placed after their first mentioning in the main text.

We have been through the manuscript and done our best to ensure that that is the case now - at least to the extent to which LaTeX allows.

L. 203: "Because ocean models approximate physical processes, they require tuning to better match observations."\_The causality is still confused, just remove this sentence. We have removed the sentence as suggested.

L. 241: "in both 2014/15 (372 m/yr to 395 m/yr, Fig. 4a) and 2015/16 (375 m/yr to 397 m/yr, Fig. 4b)" => "in both 2014/15 (minimum: 372 m yr-1, maximum: 395 m yr-1; Fig. 4a) and 2015/16 (minimum: 375 m yr-1, maximum: 397 m yr-1; Fig. 4b)"

We have changed it according to your suggestion.

Line 247: "the spatial analysis effective strain rates" => "the spatial analysis of effective strain rates"

We have changed it according to your suggestion.

Fig. 5: the channels marked in black are indiscernible from the base maps. We have changed the colors of the channels to yellow to make them more visible on the maps.

Fig. 6: what do the dashed lines in (h) and (i) denote?
Thanks for pointing this out! The dashed lines are not supposed to be there, and are leftovers from a much older version of this figure. We have made sure to remove them again.